# Single nucleosome imaging reveals loose genome chromatin networks via active RNA polymerase II

Ryosuke Nagashima[1,2]*, Kayo Hibino[1,2]*, S.S. Ashwin[3,4], Michael Babokhov[1], Shin Fujishiro[3,4], Ryosuke Imai[1,2], Tadasu Nozaki[1], Sachiko Tamura[1], Tomomi Tani[5], Hiroshi Kimura[6] ⓘ, Michael Shribak[5] ⓘ, Masato T. Kanemaki[2,7] ⓘ, Masaki Sasai[3,4] ⓘ, and Kazuhiro Maeshima[1,2] ⓘ

**Although chromatin organization and dynamics play a critical role in gene transcription, how they interplay remains unclear. To approach this issue, we investigated genome-wide chromatin behavior under various transcriptional conditions in living human cells using single-nucleosome imaging. While transcription by RNA polymerase II (RNAPII) is generally thought to need more open and dynamic chromatin, surprisingly, we found that active RNAPII globally constrains chromatin movements. RNAPII inhibition or its rapid depletion released the chromatin constraints and increased chromatin dynamics. Perturbation experiments of P-TEFb clusters, which are associated with active RNAPII, had similar results. Furthermore, chromatin mobility also increased in resting G0 cells and UV-irradiated cells, which are transcriptionally less active. Our results demonstrated that chromatin is globally stabilized by loose connections through active RNAPII, which is compatible with models of classical transcription factories or liquid droplet formation of transcription-related factors. Together with our computational modeling, we propose the existence of loose chromatin domain networks for various intra-/interchromosomal contacts via active RNAPII clusters/droplets.**

## Introduction

Genomic DNA, which encodes genetic information, is spatially and temporally organized in the cell as chromatin (Cardoso et al., 2012; Bickmore, 2013; Hübner et al., 2013; Dekker and Heard, 2015). In the process of information output (gene transcription), which specifies cellular function and subsequent fates, both chromatin organization and dynamics play a critical role in governing accessibility to genomic information. Emerging evidence reveals that the nucleosomes (10-nm fibers), consisting of genomic DNA wrapped around the core histones (Luger et al., 1997), seem to be folded rather irregularly (Eltsov et al., 2008; Fussner et al., 2012; Hsieh et al., 2015; Ricci et al., 2015; Sanborn et al., 2015; Chen et al., 2016; Maeshima et al., 2016; Ou et al., 2017; Risca et al., 2017). This implies that chromatin is less physically constrained and more dynamic than expected in the regular static structures model (Maeshima et al., 2010a). Consistently, live-cell imaging studies have long revealed a highly dynamic nature of chromatin using LacO/LacI-GFP and related systems (Marshall et al., 1997; Heun et al., 2001; Chubb et al., 2002; Levi et al., 2005; Hajjoul et al., 2013; Germier et al., 2017) and, more recently, single-nucleosome imaging (Hihara et al.,

2012; Nozaki et al., 2017) and CRISPR/dCas9-based strategies (Chen et al., 2013; Ma et al., 2016; Gu et al., 2018).

Regarding larger-scale chromatin organization, several models have been proposed, for example, chromonema fibers (Belmont and Bruce, 1994; Kireeva et al., 2004; Hu et al., 2009) or nucleosome clusters/domains (Nozaki et al., 2017) with a diameter of 100–200 nm and globular DNA replication foci/domains with an average diameter of 110–150 nm observed via fluorescent pulse labeling (Jackson and Pombo, 1998; Berezney et al., 2000; Albiez et al., 2006; Cseresnyes et al., 2009; Baddeley et al., 2010; Markaki et al., 2010; Xiang et al., 2018). Recently, chromosome conformation capture and related methods, including Hi-C (Lieberman-Aiden et al., 2009), have enabled the production of a fine contact probability map of genomic DNA and supported the formation of numerous chromatin domains, designated as topologically associating domains (Dixon et al., 2012; Nora et al., 2012; Sexton et al., 2012; Smallwood and Ren, 2013; Dekker and Heard, 2015; Nagano et al., 2017; Szabo et al., 2018), and, more recently, contact domains/loop domains (Rao et al., 2014, 2017; Eagen et al., 2015; Vian et al., 2018b), which are

[1]Genome Dynamics Laboratory, National Institute of Genetics, Research Organization of Information and Systems, Mishima, Japan; [2]Department of Genetics, School of Life Science, SOKENDAI, Mishima, Japan; [3]Department of Applied Physics, Nagoya University, Nagoya, Japan; [4]Department of Computational Science and Engineering, Nagoya University, Nagoya, Japan; [5]Eugene Bell Center for Regenerative Biology and Tissue Engineering, Marine Biological Laboratory, Woods Hole, MA; [6]Cell Biology Center, Institute of Innovative Research, Tokyo Institute of Technology, Yokohama, Japan; [7]Molecular Cell Engineering Laboratory, National Institute of Genetics, ROIS, Mishima, Japan.

*R. Nagashima and K. Hibino contributed equally to this paper; Correspondence to Kazuhiro Maeshima: kmaeshim@nig.ac.jp.

**Rockefeller University Press**
J. Cell Biol. 2019 Vol. 218 No. 5 1511–1530



https://doi.org/10.1083/jcb.201811090 1511

considered functional units of the genome with different epigenetic features. These contact probability maps have also suggested various intrachromosomal and interchromosomal domain contacts for global control of gene transcription (Dixon et al., 2012; Nora et al., 2012; Sexton et al., 2012; Smallwood and Ren, 2013; Rao et al., 2014; Dekker and Heard, 2015; Eagen et al., 2015; Nagano et al., 2017) although the underlying mechanism remains unclear.

An interesting observation, which might explain the relationship between global chromatin behavior and gene transcription, came from single-nucleosome imaging to see local chromatin movements in a whole nucleus of human cells treated with the RNA polymerase II (RNAPII) inhibitor 5,6-Dichloro-1-β-D-ribofuranosyl benzimidazole (DRB; Kwak and Lis, 2013). Contrary to the general view that transcribed chromatin regions are more open and dynamic, inhibitor treatment globally upregulated the chromatin dynamics (Nozaki et al., 2017). While recent studies reported that some specific genomic loci in human breast cancer, fly embryos, and mouse embryonic stem cells became less dynamic when actively transcribed (Ochiai et al., 2015; Germier et al., 2017; Chen et al., 2018), the transcribed chromatin regions are very limited genome-wide in human cells (Djebali et al., 2012). How then can transcription globally affect chromatin dynamics? Related to this issue, it has been long proposed that stable clusters of RNAPII work as transcription factories and immobilize chromatin to be transcribed (Buckley and Lis, 2014; Feuerborn and Cook, 2015). Recent single-molecule tracking studies have also shown that active RNAPII and other factors form dynamic clusters/droplets, possibly as a result of phase separation processes (Cisse et al., 2013; Cho et al., 2016, 2018; Boehning et al., 2018; Boija et al., 2018; Chong et al., 2018). Taken together, we hypothesized that chromatin domains form a loose network via transcription complexes for efficient gene transcription and that chromatin is globally stabilized or constrained by such a network. We inferred that inhibition or removal of RNAPII can disrupt the network connections and increase chromatin movements.

To test this hypothesis, using single-nucleosome imaging (Hihara et al., 2012; Nozaki et al., 2017), we investigated genome-wide chromatin dynamics in a whole nucleus in living human retinal pigment epithelial (RPE)-1 cells treated with various transcription inhibitors. We found that treatment with the RNAPII inhibitors, DRB, and α-amanitin (α-AM) globally raised chromatin fluctuations, suggesting fewer constraints of chromatin movement. A conditional rapid depletion of RNAPII had a similar effect. Furthermore, chromatin mobility increased in resting G0 phase cells with serum starvation and UV-irradiated cells, both of which are less transcriptionally active. Our imaging and computational modeling results suggested that chromatin is globally stabilized by loose connections through transcriptionally active RNAPII. Taken together with available data, we infer the existence of loose chromatin domain networks for various intrachromosomal and interchromosomal contacts via transient clustering of active RNAPII.

## Results

### Single-nucleosome imaging in human RPE-1 cells
We performed single-nucleosome imaging to accurately measure local chromatin dynamics in a whole nucleus and to get a clue about chromatin organization. Histone H2B was tagged with HaloTag (H2B-Halo), to which a HaloTag ligand Tetramethylrhodamine (TMR) dye can bind specifically in living cells, and the tagged H2B was stably expressed in human RPE-1 cells, an RPE cell line immortalized by hTERT (Bodnar et al., 1998; Fig. 1 A). The H2B-Halo is incorporated into the nucleosomes throughout the genome, including euchromatic and heterochromatic regions (Fig. 1 B), presumably by histone replacement on a scale of hours (Kimura and Cook, 2001). Stepwise salt washing of nuclei isolated from the established H2B-Halo–expressing cells confirmed that expressed H2B-Halo behaved similarly to endogenous H2B (Fig. S1 A), suggesting that the H2B-Halo molecules were incorporated properly into the nucleosomes of these cells. For single-nucleosome imaging, we used oblique illumination microscopy, which allowed us to illuminate a thin area within a single nucleus with reduced background noise (Fig. 1 C, green lines; Tokunaga et al., 2008; Nozaki et al., 2017). Before imaging, H2B-Halo was labeled with a low concentration of TMR (Fig. 1 D) to produce a relatively small number (∼100–200/time frame [50 ms]/nucleus) of fluorescent nucleosomes, leading to stochastic labeling of nucleosomes genome wide. Clear, well-separated dots were detected (Fig. 1 E), with a single-step photobleaching profile (Fig. S1 B), which suggested that each dot represents a single H2B-Halo-TMR molecule in a single nucleosome. The unbound free dye in the cells was negligible. The TMR dye has higher intensity and 10 times longer lifetime before photobleaching than those of photoactivatable (PA)-mCherry (Subach et al., 2009) that we previously used (Nozaki et al., 2017), both of which contribute to improved single-nucleosome imaging.

We recorded the TMR-nucleosome dots in interphase chromatin at 50 ms/frame (∼100 frames, 5 s total) in living cells (Video 1). The individual dots were fitted with a 2D Gaussian function to estimate the precise position of the nucleosome (the position determination accuracy is 15.55 nm; Fig. S1, C and D; see Materials and methods; Betzig et al., 2006; Rust et al., 2006; Selvin et al., 2007). We first tracked the movements of individual nucleosomes using u-track software (Fig. 1 F and Video 2; Jaqaman et al., 2008). Notably, we tracked only the signals of H2B-Halo-TMR incorporated into nucleosomes (Fig. 1 F and Video 2) since free histones moved too fast to detect as dots and track under our imaging conditions. Effects of nuclear movements were negligible in our conditions. From the nucleosome tracking data, we calculated mean square displacement (MSD), which shows the spatial extent of random motion in a certain time period (Dion and Gasser, 2013). The plots of calculated MSD appeared to be sub-diffusive (Fig. 1 G, black line), which is in a good agreement with those of H2B-PA-mCherry similarly expressed in RPE-1 cells (Fig. 1 G, gray line; and see also Fig. S1, E–G). Chemical fixation of the cells with formaldehyde (FA) to cross-link nucleosomes severely suppressed the movements of TMR-nucleosomes (Fig. 1 G, red line), indicating that most of the observed movement was derived from real nucleosome movements in living cells. When we analyzed the nucleosome movements within a longer tracking time, the MSD almost reached to a plateau (Fig. 1 H), which is proportional to the square of the radius of constraint (Rc; $P = 6/5 \times Rc^2$; Dion and Gasser, 2013).

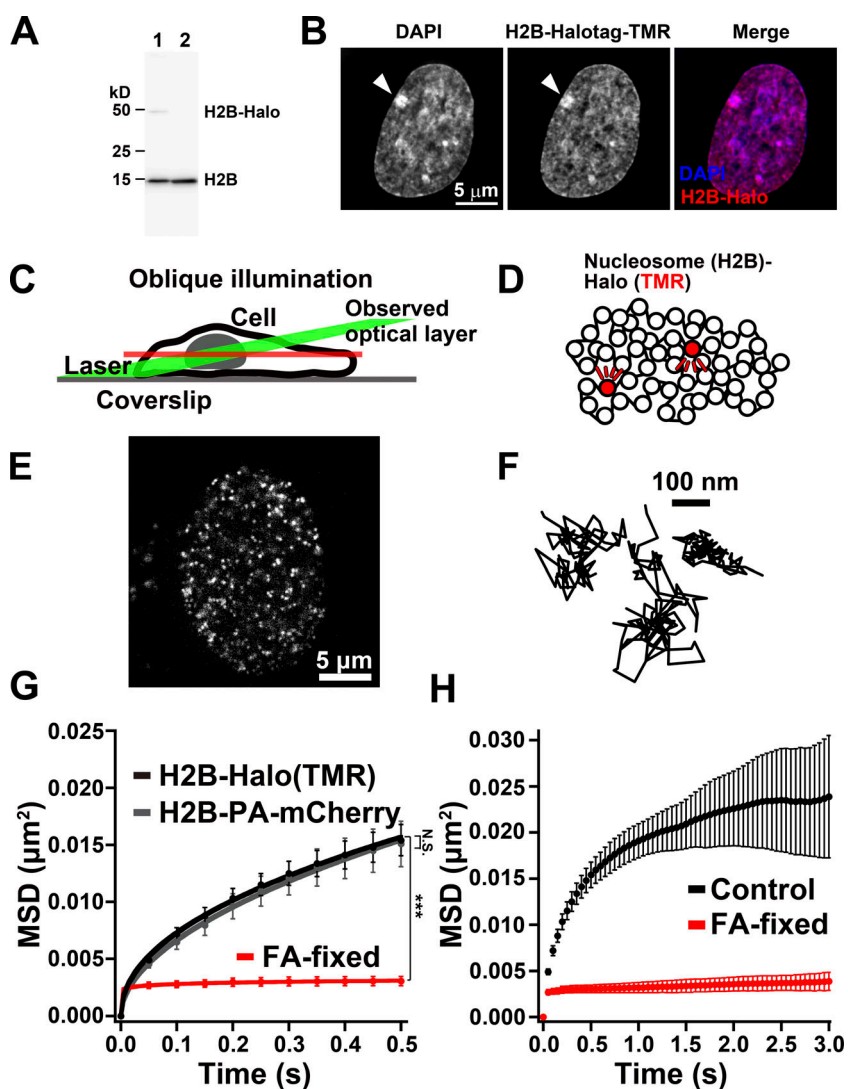

Figure 1. **Single-nucleosome imaging in living RPE-1 cells. (A)** Expression of H2B-Halo in RPE-1 cells was confirmed by Western blotting with αH2B antibody (lane 1). In lane 2, parental RPE-1 cells show no H2B-Halo signals. **(B)** RPE-1 cells expressing H2B-Halo fluorescently labeled with TMR-HaloTag ligand (center). The left panel is DNA stained with DAPI. The merged image (DNA, blue; H2B-Halo, red) is shown at right. Putative inactive X chromosome, which is highly condensed, is marked with an arrowhead. Note that the TMR labeling pattern is very similar to the DNA staining one. **(C)** Scheme of oblique illumination microscopy. This illumination laser (green) can excite fluorescent molecules within a limited thin optical layer (red) of the nucleus and reduce background noise. **(D)** A small fraction of H2B-Halo was fluorescently labeled with TMR-HaloTag ligand (red). The labeled nucleosome movements can be tracked at super-resolution. **(E)** A single-nucleosome (H2B-Halo-TMR) image of a living RPE-1 nucleus after background subtraction. **(F)** Representative three trajectories of the tracked single nucleosomes. **(G)** MSD plots (±SD among cells) of single nucleosomes in living interphase (black) and FA-fixed (red) RPE-1 cells from 0.05 to 0.5 s. For comparison, MSD plots of single nucleosomes labeled with PA-mCherry (H2B-PA-mCherry) in living interphase RPE-1 cells (gray) are also shown. For each sample, $n$ = 20–25 cells. N.S. (not significant, P = 0.47) and ***, P < 0.0001 (P = 1.5 × 10$^{-11}$) for H2B-Halo versus FA-fixed cells by the Kolmogorov–Smirnov test. **(H)** MSD plots (±SD among cells) of single nucleosomes in living (black) and FA-fixed (red) RPE-1 cells in a longer tracking time range from 0.05 to 3 s. For each sample, $n$ = 20 cells. In the MSD analyses for single nucleosomes, the originally calculated MSD was in two dimensions. To obtain 3D values, the original values of MSD were multiplied by 1.5 (4 to 6 Dt). The plots were fitted as a subdiffusive curve: MSD = 0.018t$^{0.28}$ in a living cell; MSD = 0.003t$^{0.01}$ in an FA-fixed cell. Rc (estimated radius of constraint of the nucleosome motion), 141 ± 19.2 nm (mean ± SD) in living cells; 56 ± 6.7 nm in FA-fixed cells. Their Rc values are significantly different: P = 2.2 × 10$^{-10}$ by the Kolmogorov-Smirnov test.

The estimated radius of constraint of the nucleosome motion in living cells is 141 ± 19.2 nm (mean ± SD) while that of nucleosomes in FA-fixed cells is 56 ± 6.7 nm. Spatial distributions of the obtained nucleosome movements were also visualized as a "chromatin heat map" in the nucleus: larger chromatin movement appeared as more red (or hot), and smaller movement appeared as more blue (or cold) pixels (Fig. S1 H).

### Transcriptional inhibition by DRB or α-AM diminished constraints of local chromatin movements

To examine the role of the transcriptional process in the constraint of chromatin motion, we first treated cells with DRB or α-AM (Bensaude, 2011; Kwak and Lis, 2013). DRB is a selective inhibitor of CDK9 kinase in the P-TEFb complex that phosphorylates Ser2 in the C-terminal domain (CTD) of the largest subunit of RNAPII, RPB1 in eukaryotic cells and inhibits the transcription elongation process (Fig. 2 A; Bensaude, 2011; Kwak and Lis, 2013). It was also reported that DRB caused dissociation of the RNAPII elongation complex from chromatin (Kimura

et al., 2002). α-AM binds with high specificity and high affinity ($K_d$ = 3–4 nM) near the catalytic active site of RNAPII (Bushnell et al., 2002) and induces degradation of the RPB1 (Nguyen et al., 1996). Immunostaining of two active RNAPII markers, phosphorylated serine 5 (Ser5P) and serine 2 (Ser2P) of the CTD (Stasevich et al., 2014), which are known to be involved in initiation and elongation process of RNAPII, respectively (Fig. 2 A), showed punctate foci throughout the nucleoplasm in the untreated control (Fig. 2, B and C; and Fig. S2 A). Both inhibitors treatments significantly reduced the amount of the two active RNAPII marks (Fig. 2, B and C; and Fig. S2 A). Consistently, these treatments markedly suppressed global RNA synthesis in the cells, which was measured by incorporation of a ribonucleotide analog, 5-ethynyl uridine (EU; Fig. 2 D). In these cells treated with DRB or α-AM, the local chromatin movements were globally up-regulated in a dose-dependent manner (Fig. 3 A; Video 3; and Fig. S2, B and C), consistent with the chromatin heat map (Fig. 3 B) and our previous report (Nozaki et al., 2017). Interestingly, when the effects of DRB and α-AM were compared, we found that the amount of RNAPII-Ser5P seems to be

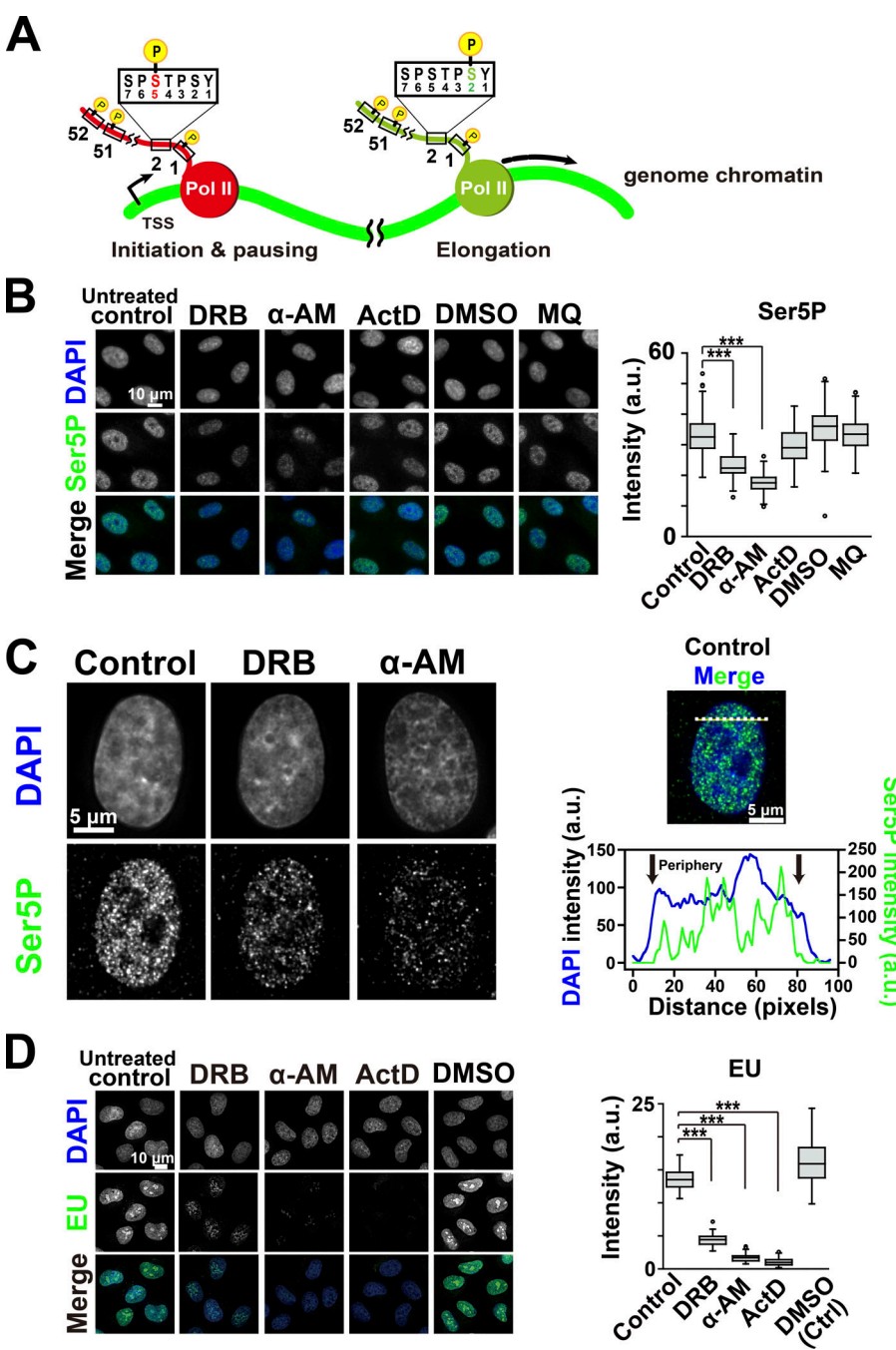

Figure 2. **Decrease in the amount of active RNAPII and RNA synthesis by RNAPII inhibitors.** **(A)** Scheme for RNAPII regulation by phosphorylation of its CTD repeats, (YSPTSPS) × 52. In the initiation process, RNAPII, in which Ser5 of CTD is phosphorylated, stays around the initiation site (red, RNAPII-Ser5P) on the template DNA. For elongation, with phosphorylation of Ser2 of CTD, the RNAPII complex goes along the template DNA (green, RNAPII-Ser2P). Note that the scheme is highly simplified. **(B)** Left: Effect of RNAPII inhibitors on RNAPII activity. RNAPII activity in RPE-1 cells was monitored by immunostaining of the active RNAPII marker (Stasevich et al., 2014), Ser5P, of RNAPII CTD. The inhibitors used were α-AM, DRB, and ActD. As solvent controls, DMSO and ultrapure water (MQ) were used. First row, DNA staining with DAPI; second row, immunostaining of Ser5P of RNAPII CTD; third row, merged images. Right: The quantification of RNAPII-Ser5P signal intensity is shown as a box plot. The median intensities of Ser5P: 32.6 ($n$ = 118 cells) in control; 22.4 ($n$ = 121 cells) in DRB; 17.5 ($n$ = 141 cells) in α-AM; 29.0 ($n$ = 130 cells) in ActD. ***, P < 0.0001 by the Wilcoxon rank sum test for control versus DRB (P < 2.2 × 10$^{-16}$), and for control versus α-AM (P < 2.2 × 10$^{-16}$). **(C)** Left: Active RNAPII (Ser5P) distribution in RPE-1 cells with DRB or α-AM or without inhibitors (untreated control) observed by immunostaining. Typical images with deconvolution by DeltaVision Softworx software are shown. RNAPII-Ser5P formed clusters and distributed in the nucleoplasm except for nuclear periphery and nucleoli. Right: The intensity line profiles (bottom) of DAPI (blue) and RNAPII-Ser5P (green) on the dotted line in the merged image (top) show that the nuclear periphery regions (arrows) are quite free from RNAPII-Ser5P signals. **(D)** Left: Verification of RNA synthesis inhibition by RNAPII inhibitors (α-AM, DRB, and ActD) with EU incorporation into RNA. The incorporated EU was detected with Alexa Fluor 594–labeling by click chemistry. For each condition, $n$ = 45–53 cells. Right: Box plot of EU signal intensity. The median intensities of EU are 13.5 ($n$ = 49 cells) in control, 4.44 ($n$ = 48 cells) in DRB, 1.69 ($n$ = 45 cells) in α-AM, 0.975 ($n$ = 45 cells) in ActD, and 16.1 ($n$ = 53 cells) in DMSO. Note that the inhibitor treatments decreased RNA transcription. ***, P < 0.0001 by the Wilcoxon rank sum test for control versus DRB (P < 2.2 × 10$^{-16}$), for control versus α-AM (P < 2.2 × 10$^{-16}$), and for control versus ActD (P < 2.2 × 10$^{-16}$).

well correlated with constraints of chromatin movements (Fig. 2, B and C; and Fig. 3 A). Using MSD analysis with a longer tracking time (Fig. 3 C), we estimated that Rc of the nucleosome motion in the cells treated with DRB and α-AM increased from 141 ± 19.2 nm (control) to 149 ± 20.4 nm and 164 ± 22.0 nm, respectively, suggesting a reduction in constraints. On the other hand, an RNA polymerase I Inhibitor CX-5461, which suppressed rRNA synthesis in nucleoli (Fig. 4 A; Drygin et al., 2011), but not the amount of RNAPII-Ser5P (Fig. 4 B), had almost no effect on

the local chromatin dynamics (Fig. 4 C). Furthermore, inhibition of the large RNA splicing complex, which is associated with active RNAPII during its elongation process, by Pladienolide B (Koga et al., 2015) only slightly affected the chromatin dynamics (Fig. 4 D) although pre-mRNA production was almost normal and subsequent splicing processes were severely inhibited by the inhibitor treatment (Fig. 4 E). Taken together, these results suggested a specific role of the active RNAPII in restricting chromatin dynamics.

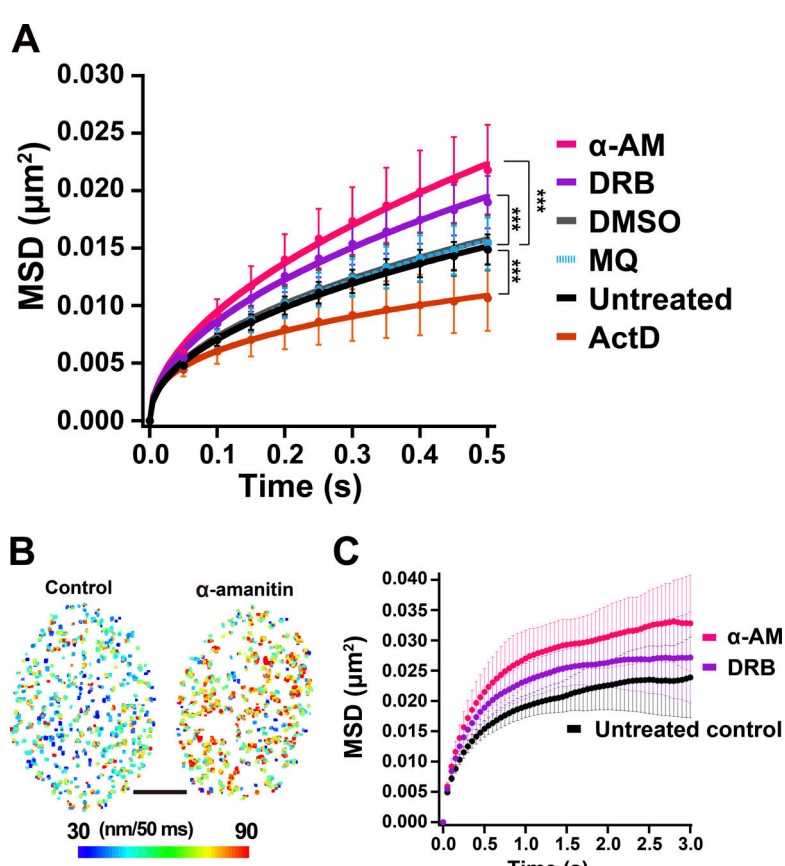

**A**

**B** Control    α-amanitin

30 (nm/50 ms) 90

**C**

Figure 3. **Increased chromatin dynamics by RNAPII inhibitors. (A)** MSD plots (±SD among cells) of nucleosomes in the RPE-1 cells treated with RNAPII inhibitors, α-AM (pink), DRB (purple), and ActD (brown). The controls are DMSO (gray), MQ (light blue), or untreated (black). For each condition, n = 20 cells. Note that the inhibition of RNAPII increased the chromatin dynamics, except for ActD. ***, P < 0.0001 by the Kolmogorov–Smirnov test for untreated control versus DRB (P = 1.4 × 10⁻⁷), for untreated control versus α-AM (P = 1.0 × 10⁻⁸), and for untreated control versus ActD (P = 9.5 × 10⁻⁶). **(B)** Chromatin heat maps of the nuclei treated with (right) and without α-AM (left): Larger chromatin movement appears as more red (or hot), and smaller movement appears as more blue (or cold) pixels. Note that the heat map of the nucleus with α-AM turned more red. Bar, 5 µm. **(C)** MSD plots (±SD among cells) of single nucleosomes in RPE-1 cells treated with RNAPII inhibitor (α-AM, DRB) or without inhibitors (control) from 0.05 to 3 s. For each sample, n = 20 cells. The inhibitor treatments increased chromatin dynamics with less constraint. The plots were fitted as a subdiffusive curve: MSD = 0.018t⁰·²⁸ in untreated cells; MSD = 0.022t⁰·²⁶ in DRB-treated cells; MSD = 0.025t⁰·²⁸ in α-AM–treated cells. Rc: 141 ± 19.2 nm in untreated cells, 149 ± 20.4 nm in DRB-treated cells, and 164 ± 22.0 nm in α-AM–treated cells. Rc values between untreated control and α-AM–treated cells are significantly different: P = 0.018 by the Kolmogorov-Smirnov test.

Since the DRB and α-AM treatments described above reduced the amount of active RNAPII on chromatin, we wondered whether more stable RNAPII binding to chromatin might induce chromatin stabilization and constraint. To test this possibility, we examined the effect of another inhibitor, actinomycin D (ActD), which induces stalling of active RNAPII on chromatin (Kimura et al., 2002). While ActD treatment reduced global RNA synthesis in the cells (Fig. 2 D), the amounts of both active RNAPII marks in the treated cells were similar or slightly higher than those of untreated control cells (Fig. 2 B and Fig. S2 A), suggesting more active RNAPII on the chromatin. As contrasted with DRB and α-AM, ActD treatment decreased the chromatin dynamics and induced more constraints (Fig. 3 A). This result supports the notion that active RNAPII complexes on chromatin constrained the local chromatin movements throughout the whole genome.

As shown by EU incorporation, RNAPII inhibition greatly decreased RNA production in the nuclei (Fig. 2 D). To exclude the possibility that transcription inhibition might affect the physical chromatin environment and subsequently change chromatin behavior, we further examined two factors, total material density (molecular crowding) and free Mg²⁺ concentration in the nuclei. Total densities in the nucleoplasm of α-AM–treated and untreated RPE-1 cells, which were directly measured by orientation-independent differential interference contrast (OI-DIC) microscopy (for the method details, see Fig. S2 E, Materials and methods, and Imai et al., 2017), were both

~130 mg/ml (Fig. S2 D). Although free Mg²⁺ can greatly condense chromatin in the cell by neutralizing the electrostatic charge of DNA (Hansen, 2002; Maeshima et al., 2018), we found no significant difference in free Mg²⁺ levels between the treated and untreated cells (Fig. S2 F). Again, the results strengthened the notion that active RNAPII globally restricts chromatin dynamics.

If chromatin movements were globally constrained by transcriptionally active RNAPII, we inferred that the transcriptional inhibition should not affect chromatin dynamics in heterochromatin regions. To test this possibility, we focused on the nuclear bottom surfaces (nuclear periphery; Shinkai et al., 2016), where condensed lamina-associated domains (van Steensel and Belmont, 2017) are enriched. The regions have fewer active-RNAPII (Fig. 2 C, right; Boettiger et al., 2016), and their chromatin is less mobile (Nozaki et al., 2017). To visualize the nuclear surfaces (nuclear periphery), we adjusted the angle of laser illumination to efficiently capture a nuclear membrane marker, NUP107-Venus (Maeshima et al., 2010b; Fig. S2 G). As reported previously (Shinkai et al., 2016; Nozaki et al., 2017), chromatin dynamics on the nuclear surface were lower (Fig. 4 F and Video 4) than those of nuclear interior (Video 1), which contained more euchromatin regions. α-AM treatment, which showed a strong effect in the nuclear interior, did not increase the chromatin dynamics on the nuclear surfaces (nuclear periphery; Fig. 4 F). The obtained finding suggests that RNAPII activity only affected the chromatin dynamics of actively transcribed regions.

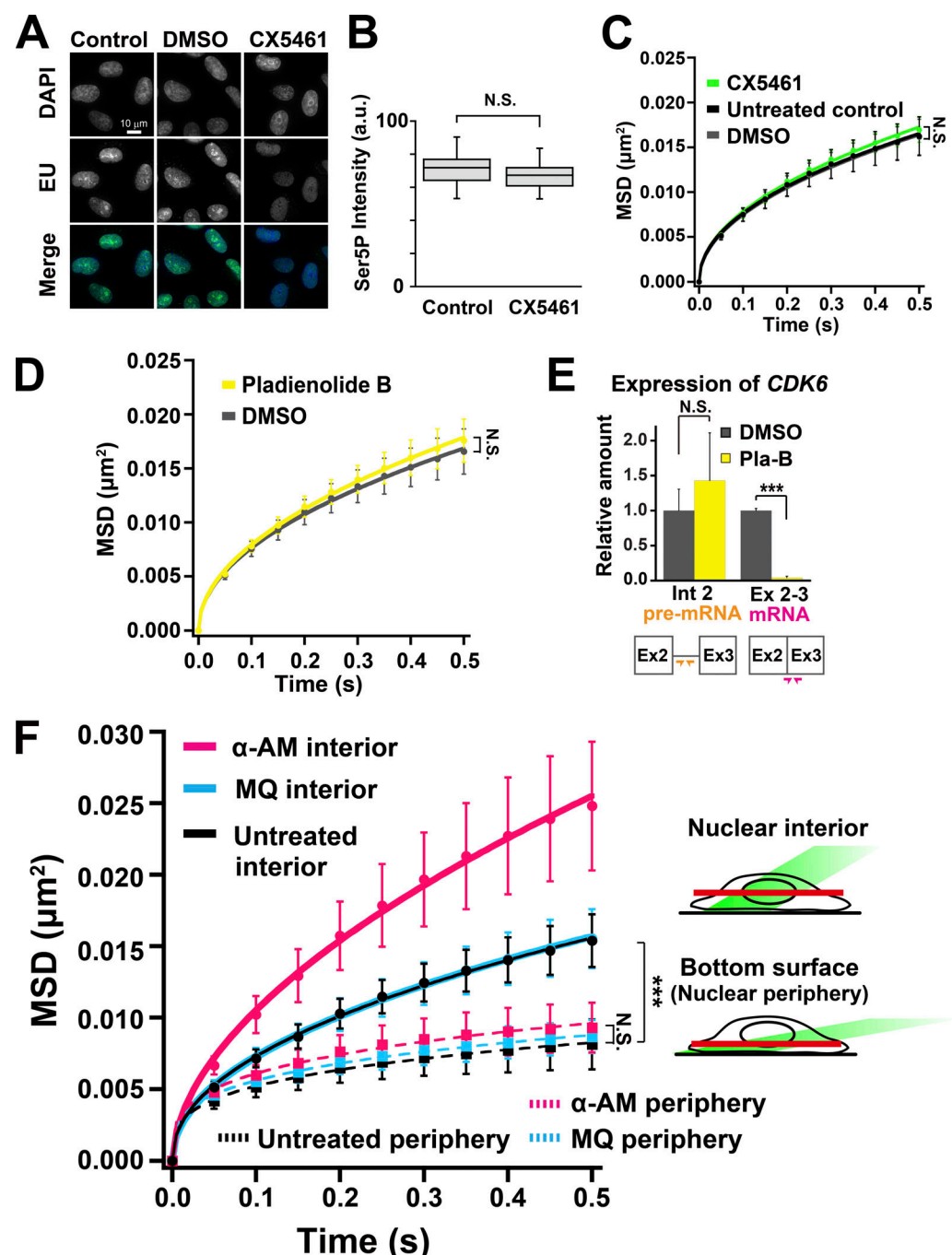

Figure 4. **Inhibitors of RNA polymerase I and splicing had little influence on the chromatin dynamics. (A)** Verification of RNA polymerase I inhibition in RPE-1 cells with CX5461 by EU incorporation. **(B)** The box plots show RNAPII-Ser5P signal intensity in control and CX5461-treated RPE-1 cells. The median intensities of Ser5P are 71.8 in control ($n$ = 30 cells) and 67.3 in CX5461 ($n$ = 30 cells). N.S. (P = 0.066) by the Wilcoxon rank sum test. **(C)** MSD plots (±SD among cells) of single nucleosomes in RPE-1 cells treated with polymerase I (RNAPI) inhibitor (CX5461, green), solvent (DMSO, gray), or none (control, black). For each condition, $n$ = 37–39 cells. Note that the effect of RNAPI inhibition on the chromatin dynamics is very small. N.S. (P = 0.40) by the Kolmogorov–Smirnov test for untreated control versus CX5461. **(D)** MSD plots (±SD among cells) of single nucleosomes in RPE-1 cells treated with splicing inhibitor, Pladienolide B (Pla-B, yellow) or DMSO (gray). For each condition, $n$ = 20 cells. N.S. (P = 0.34) by the Kolmogorov–Smirnov test. **(E)** Verification of splicing inhibition in RPE-1 cells treated with Pla-B by quantitative RT-PCR. Relative amounts of spliced (Ex 2–3 mRNA) and nonspliced (Int 2 premRNA) CDK6 RNA products in RPE-1 cells treated with Pla-B (yellow) or DMSO (gray) are shown. Schematic representation of primer positions for Ex 2–3 mRNA (pink arrows) and Int 2 pre-mRNA (orange arrows) are also shown at the bottom. Averaged relative amounts of the products were shown with SD ($n$ = 3). N.S. (P = 0.38) and ***, P < 0.0001 (P = $1.7 \times 10^{-6}$) by Student's $t$ test. **(F)** Left: MSD plots (±SD among cells) of single nucleosomes on the interior and peripheral layers of the RPE-1 nuclei treated with RNAPII inhibitor, α-AM (pink), solvent (MQ, light blue), or none (control, black). For each condition, $n$ = 15 cells. Note that on the nuclear periphery, the chromatin dynamics were not significantly affected by α-AM treatment. N.S. (P = 0.075) for control periphery versus α-AM periphery and ***, P < 0.0001 for control periphery versus control interior (P = $3.9 \times 10^{-7}$) by the Kolmogorov–Smirnov test. There was no significant difference between the MSD of α-AM–treated interior in Fig. 4 F and that of α-AM–treated in Fig. 3 A (P = 0.13). Right: Schematic representation for nuclear interior (top) and periphery (bottom) imaging.

## Rapid depletion of RNAPII by auxin-inducible degron system diminished the chromatin constraints

Inhibitor treatments always have a risk for various indirect effects. For instance, it was reported that long DRB treatment could disrupt nucleolar structure and disperse them throughout the nuclear interior (Chubb et al., 2002). To validate the involvement of RNAPII in the constraint of chromatin movements more directly, we generated cells that enable us to perform rapid and specific degradation of RPB1, the largest subunit of RNAPII, by an auxin-inducible degron (AID) system (Fig. 5 A; Natsume et al., 2016; Yesbolatova et al., 2019 Preprint). Using CRISPR/Cas9-based genome editing, we introduced a cassette encoding mini-AID (mAID) and fluorescent protein mClover (mAID+mClover) at the initiation site of the endogenous RNAPII gene locus (POLR2A; Fig. 5 B) in human colon adenocarcinoma DLD-1 cells expressing OsTIR1, which is involved in the induced degradation process (Fig. 5 A; Natsume et al., 2016; Yesbolatova et al., 2019 Preprint). The DLD-1 cells that carried a proper insertion of mAID+mClover tag into POLR2A gene were selected with hygromycin resistance and auxin sensitivity and were further confirmed by PCR to obtain bi-allelic–tagged RPB1 (Fig. 5 C). We also verified with immunoblotting that all fractions of RPB1 protein in clones 1 and 5 were rapidly degraded upon auxin addition (Fig. 5 D). For chromatin dynamics analysis, H2B-Halo was introduced into clone 5, and we established a cell line stably expressing H2B-Halo in the clone 5 background (Fig. 5 E, untreated). In the established cells, RPB1 was depleted within 1 h after auxin addition (Fig. 5 E, +Auxin). Similar to the observation when treated with DRB or α-AM, rapid depletion of RNAPII significantly increased chromatin dynamics (Fig. 5 F). Furthermore, 12 h after washing out auxin, the amount of RNAPII and chromatin dynamics returned to the untreated control levels (Fig. 5, E and F). The rapid depletion and rescue experiments support our hypothesis that RNAPII directly constrains chromatin movements.

To confirm the above notion in a different cell line, we generated analogous cells in the human HCT116 background (Natsume et al., 2016; Yesbolatova et al., 2019 Preprint) using the same strategy as described above (Fig. S3). In the HCT116 background, RPB1 was again rapidly degraded upon auxin addition (Fig. S3, A and B). The rapid depletion of RNAPII increased chromatin dynamics and suppressed chromatin constraints (Fig. S3 C), demonstrating that the effect of RNAPII depletion is consistent across different cell types.

## Decreased chromatin constraints in the resting G0 state of RPE-1 cells

To investigate chromatin constraints by active RNAPII in a more physiological state, we induced RPE-1 cells into a resting and transcriptionally less active G0 state by serum removal from the culture medium (Fig. 6 A). G0 entry was confirmed by the loss of the proliferation markers Ki67 and TopoisomeraseIIα (TopoIIα) in the treated cells (Fig. S4, A and B). Serum starvation for 3 d induced 70% of the cells into G0, and almost all cells were in G0 after 7 d of starvation. Depending on the starvation periods, the chromatin dynamics increased in the resting G0 cells (Fig. 6 B). Consistently, signals of two active RNAPII markers (Stasevich

et al., 2014) significantly decreased in the resting G0 cells (Fig. 6 C and Fig. S4 C). Notably, the decrease in RNAPII-Ser5P, rather than RNAPII-Ser2P, was well correlated with an increase in local chromatin dynamics, i.e., a decrease in constraints of local chromatin movements (Fig. 6 B). To exclude the possibility that serum starvation changes global chromatin organization to induce an uneven TMR-labeling, which prefers open chromatin regions, we performed TMR-labeling before serum starvation and obtained a consistent result (Fig. S4 D).

We then examined the effect of serum re-addition on resting G0-cells. One day after serum restoration, the cells became Ki67-positive (Fig. S5 A) and suppressed chromatin movements (Fig. 6 E), which are similar to those of normal proliferating cells. Concurring with this dynamics decrease, the two active RNAPII marks were restored (Fig. 6 D and Fig. S5 B). Taken together, the decrease in transcription in the resting G0-cells suppressed constraints of chromatin movements.

## Transcription inhibition in response to UV irradiation decreased the chromatin constraints

Next, we pursued chromatin dynamics in another physiological state: the inducible suppression of transcription by DNA damage. Although DNA damage in transcribed regions is efficiently repaired by transcription-coupled nucleotide excision repair, if this fails, RNAPII is thought to be degraded by the ubiquitin-proteasome system (Wilson et al., 2013). When RPE-1 cells were exposed to UV, the active RNAPII signals decreased in a dose-dependent manner (Fig. 7 A and Fig. S5 C). Transcription inhibition in response to UV irradiation increased the chromatin movements and decreased constraints, coinciding with the irradiated UV dose (Fig. 7 B). This supports the notion of chromatin constraints via active RNAPII even in physiological contexts, although we cannot fully rule out the possibility that DNA damage also contributes to chromatin decondensation and subsequent increase in its motion (Dellaire et al., 2006).

## Loose genome chromatin domain network via active RNAPII

What is the underlying molecular mechanism for globally constraining chromatin motion? On the basis of available and obtained data, we hypothesize that transcription complexes/clusters including RNAPII-Ser5P weakly connect multiple chromatin domains into a loose spatial genome chromatin network (Fig. 8 A) and that, thereby, this loose network globally stabilizes or constrains chromatin (Fig. 8 A). Recent studies have shown that active RNAPII, Mediator, and other transcription factors form dynamic clusters/droplets, presumably by a phase separation process (Cisse et al., 2013; Cho et al., 2016, 2018; Boehning et al., 2018; Boija et al., 2018; Chong et al., 2018; Sabari et al., 2018). Other studies reported that the P-TEFb complex consisting of Cyclin T1 and CDK9 kinase, which interacts with RNAPII and phosphorylates its CTD, forms a number of dynamic clusters/droplets in living cells (Ghamari et al., 2013; Lu et al., 2018). These P-TEFb clusters/droplets can provide multiple weak interactions between RNAPII-Ser5P and DNA, especially considering that CDK9 and RNAPII-Ser5P co-occupy thousands of promoter-proximal regions of transcribed genes in the reported genomics data (Ghamari et al., 2013; Fig. 8 A). In this

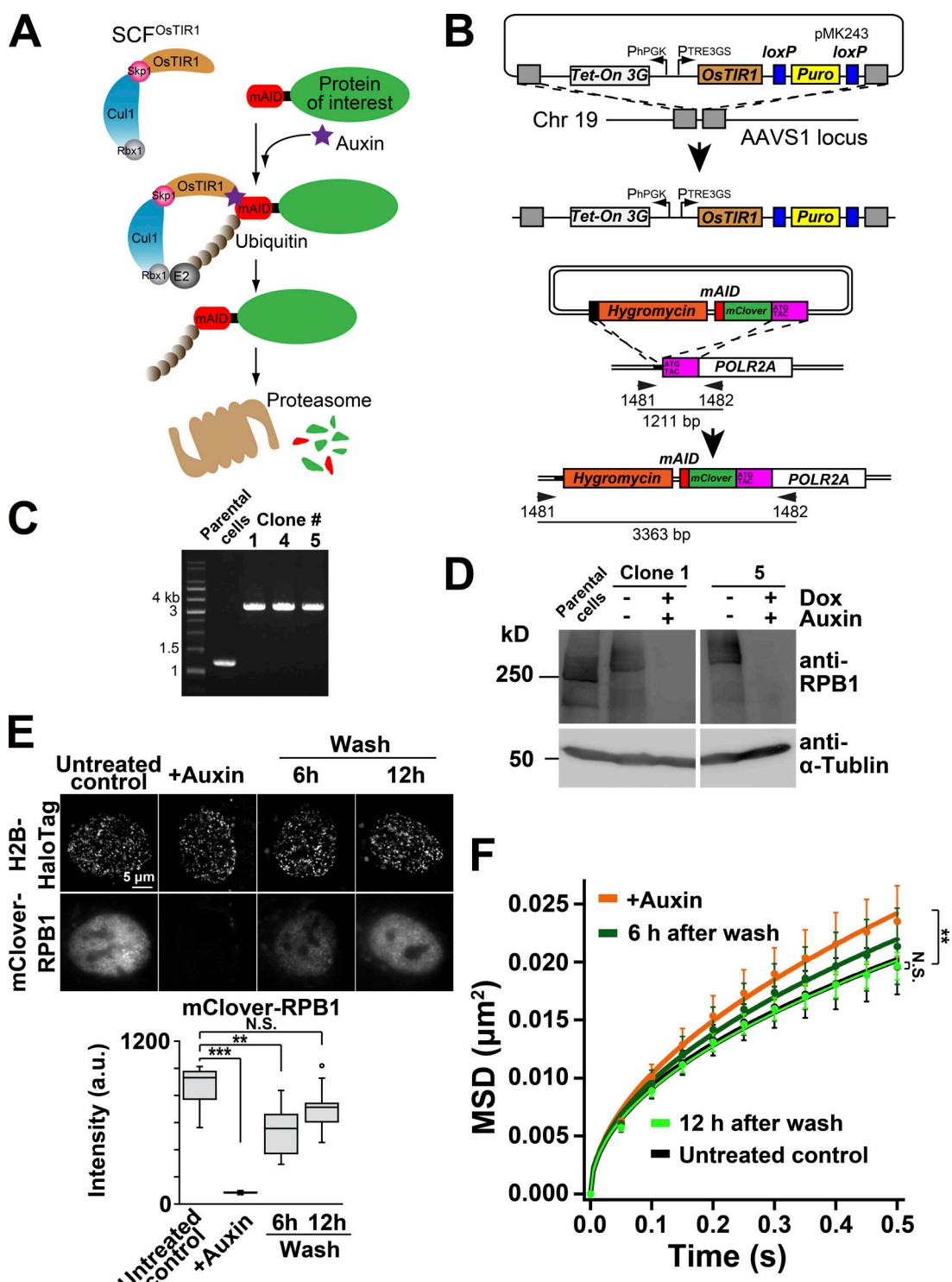

Figure 5. **Rapid degradation of RNAPII increased the chromatin dynamics. (A)** A schematic illustration of the AID system (Natsume et al., 2016; Yesbolatova et al., 2019 *Preprint*). OsTIR1, which was expressed by addition of doxycycline, can form a functional SCF^OsTIR1 E3 ligase complex with the endogenous components in human cells. In the presence of auxin, a protein of interest fused with mAID is rapidly degraded upon polyubiquitylation. **(B)** Experimental scheme used to introduce Tet-OsTIR1 at the safe-harbor AAVS1 locus in human colorectal carcinoma DLD-1 cells (top) and used to generate mAID-mClover-RPB1 (mAC-RPB1) cells (bottom) by a CRISPR/Cas9 genome editing method. Genomic PCR to test the genotype of clones after hygromycin selection was performed. Primer sets and expected PCR products are shown in B (bottom). After integration at the *POLR2A* gene encoding the largest subunit of RNAPII, RPB1, the PCR primers should give rise to ~3.4-kb products in DLD-1 cells. **(C)** PCR confirmed that both alleles of *POLR2A* gene were tagged with mAID-mClover. **(D)** RNAPII degradation in DLD-1 cells (Clone 1 and Clone 5) after auxin treatment was verified by immunoblotting by using an antibody against the RPB1 CTD. α-Tubulin antibody was used as a control. Since the RPB1 in the AID cells was fused with mAID and mClover (totally ~35 kD), the detected RPB1 in Clone 1 and Clone 5 has a slightly larger size than that of the parental cells. Note that the auxin treatment induced RNAPII degradation. **(E)** Fluorescent images of H2B-Halo-TMR (top) and mClover-RPB1 (middle) in living DLD-1 cells: From left to right, the cells before (untreated control) and after treatment with auxin

for 1 h (+Auxin), the cells incubated for 6 h and 12 h after washing out auxin. Bottom: The median intensities of mClover-RPB1 in the indicated cells are the following: 931 ($n$ = 10) in untreated control; 78.8 ($n$ = 10) in +Auxin; 557 ($n$ = 10) at 6 h after washing; 713 ($n$ = 10) at 12 h after washing cells. ***, P < 0.0001 (P = 1.1 × 10$^{-5}$), **, P < 0.01 (P = 7.2 × 10$^{-4}$), and N.S. (P = 0.063) by the Wilcoxon rank sum test. **(F)** MSD plots (±SD among cells) of nucleosomes in DLD-1 cells with indicated conditions: The cells before (untreated control, black) and after treatment with auxin for 1 h (+Auxin, orange); after washing out auxin, the cells incubated for 6 h (dark green) and 12 h (light green). For each condition, $n$ = 20 cells. The prompt degradation of RNAPII increased the chromatin dynamics. Note that DLD-1 cells have generally higher MSD values than RPE-1 cells due to unknown reasons. **, P < 0.01 for control versus +Auxin (P = 2.7 × 10$^{-4}$) and N.S. (P = 0.83) by the Kolmogorov–Smirnov test.

context, the P-TEFb clusters and RNAPII-Ser5P can work as "hubs" and "glues" for the multiple weak interactions in the network, respectively (Fig. 8 A). The chromatin domain movements, which seem to be driven essentially by Brownian motion in living cells (Nozaki et al., 2017), are thus globally constrained by the loose network (Fig. 8 A). Consistent with this hypothesis, knockdown (KD) of CDK9 kinase by siRNA up-regulated chromatin movements upon reduction of CDK9 protein levels (Fig. 8, B and C). Furthermore, DRB treatment, which is also a known inhibitor of P-TEFb (Bensaude, 2011), had a similar result (Fig. 3 A). These data suggested that perturbations of P-TEFb clusters/droplets lead to loss of the chromatin network hubs and subsequent increase in chromatin movements.

We inferred that inhibition or removal of RNAPII-Ser5P can loosen network connections and increase chromatin movements. To test this idea, we performed a Brownian dynamics simulation to reconstruct a chromatin environment (Fig. 9 A). We put four dynamic polymer chains and four hubs (P-TEFb clusters) in a cubic box (1.5 µm each side; for details, see Materials and methods). Each chain has 80 beads (Fig. 9 A, green spheres) connected by invisible springs, each of which corresponds to a chromatin domain with 100 kb. In this system, the glue (RNAPII-Ser5P; Fig. 9 A, red spheres) can weakly bind to the hubs (the P-TEFb clusters; Fig. 9 A, pink spheres) and mediate transient interactions between the hubs and beads (green spheres, chromatin domains). Note that among the existing beads only a few can interact with the hub through the glues (Fig. 9 A), which mimics the limited genome-wide transcription of chromatin regions (Djebali et al., 2012). All bead movements were tracked and analyzed by changing the number of glues (Fig. 9 B). As we expected, the addition of glues decreased the beads movements (Fig. 9 B and Videos 5 and 6). The MSD distribution plots of total beads show that a slow bead fraction (Fig. 9 C, arrow), which is constrained by the hubs and glues, lowered the ensemble average of total bead dynamics in the system (Fig. 9 B). This computational modeling data supports our notion that chromatin forms a loose network via RNAPII-Ser5P and that thereby chromatin is globally constrained by the network (Fig. 8 A).

## Discussion

Using single-nucleosome imaging and computational modeling, we investigated genome-wide chromatin dynamics in a whole nucleus of living cells, and demonstrated the constraints on chromatin movements via active RNAPII (Fig. 8 A). Similar constraints by active RNAPII were observed in more physiological contexts of the cell, such as the resting G0-state with serum starvation (Fig. 6, B and E) or the UV-irradiated state

(Fig. 7 B). Since the general view is that transcribed chromatin regions are more open and dynamic, the constraining role of active RNAPII was an unexpected finding but seems to be a general response in the cells. In contrast, behavior of the chromatin around the nuclear periphery (surface) is insensitive to transcriptional suppression presumably because these regions are enriched with lamina-associated domains (van Steensel and Belmont, 2017) tethered to inner nuclear membrane proteins (Lemaître and Bickmore, 2015) and have few active-RNAPII (Fig. 2 C). Our finding suggests that while heterochromatin, whose chromatin is cross-linked by protein factors such as lamina or possibly HP1, is less mobile, motion of actively transcribed regions in euchromatin is also constrained by transcriptional machinery, including active RNAPII. We would like to emphasize the importance of seeing not only the individual "tree" (genome locus) but also the "forest" (genome-wide chromatin) to understand the nature of chromatin organization in the living cells.

Our finding demonstrates that RNAPII is directly involved in constraining genome chromatin and also suggests the existence of a loose chromatin domain network in a whole nucleus via RNAPII-Ser5P (Fig. 8 A). We cannot completely rule out the possibility that the observed constraints in chromatin motion by transcription could be a result of increased local stiffness upon RNAPII binding. This possibility is, however, unlikely because the amount of RNAPII-Ser5P, rather than RNAPII-Ser2P, is well correlated with constraints of chromatin movements (Figs. 2, 3, 6, and 7): lower RNAPII-Ser5P levels lead to fewer chromatin constraints. RNAPII-Ser2P, which is involved in transcription elongation and RNA processing (Fig. 2 A), appeared to be less relevant. Furthermore, inhibition of the large splicing complex, which is supposedly associated with the RNAPII-Ser2P complex going along the template DNA, did not affect the local chromatin motion (Fig. 4 D). From this evidence it is apparent that the potential stiffening effect of RNAPII binding to chromatin is alone not sufficient to explain the constraining effects that we observed. The formation of chromatin-constraining transcription hubs fits well with available evidence and can explain why active RNAPII, which locates only at limited regions of the genome (Djebali et al., 2012), can globally constrain chromatin motion, as discussed below.

First, our model (Fig. 8 A) that P-TEFb clusters and RNAPII-Ser5P are a hub and glue, respectively, is supported by the perturbations of P-TEFb clusters by DRB (Fig. 3 A) or CDK9 KD (Fig. 8 B) or removal of RNAPII (glue; Fig. 5 F and Fig. S3 C), which can potentially remove connections between the hubs and chromatin domains. These manipulations all resulted in a decrease in global chromatin constraints. Our computational simulation showed that a slow chromatin fraction, which is

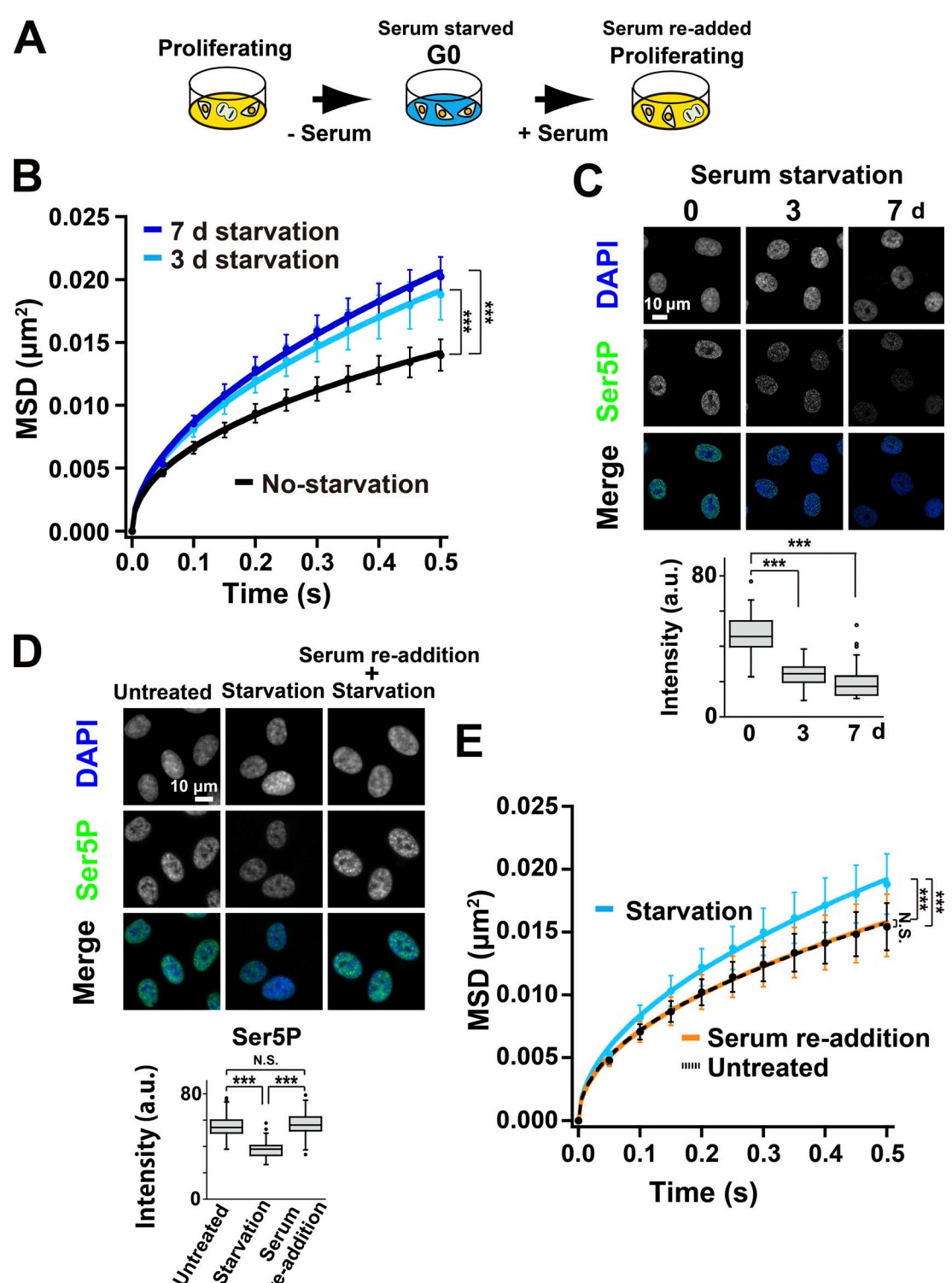

Figure 6. **Serum starvation increased chromatin dynamics. (A)** Experimental scheme. Proliferating cells were starved by removing serum from the culture medium. Most of the cells entered the quiescent G0 phase. The starved cells were then stimulated with serum re-addition to re-enter them into the proliferating state. **(B)** MSD plots (±SD among cells) of nucleosomes in RPE-1 cells (black) and with the serum starvation for 3 d (light blue) or 7 d (dark blue). For each condition, $n$ = 20 cells. Note that the chromatin dynamics increased depending on the serum starvation period. ***, P < 0.0001 by the Kolmogorov–Smirnov test for no starvation versus 3-d starvation (P = $1.1 \times 10^{-8}$) and for no starvation versus 7-d starvation (P = $1.5 \times 10^{-11}$). **(C)** Top: Verification of RNAPII activity of RPE-1 cells without (0 d) or with the serum starvation for 3 d and 7 d by immunostaining of Ser5P of the RPB1 CTD in RNAPII. Bottom: Quantifications of RNAPII Ser5P signal intensity are shown as box plots. The median intensities of Ser5P are 45.5 ($n$ = 38) at 0 d starvation, 24.4 ($n$ = 47) at 3 d, and 17.3 ($n$ = 42) at 7 d. RNAPII activity decreased after serum starvation. ***, P < 0.0001 by the Wilcoxon rank sum test for 0 d versus 3 d (P < $2.2 \times 10^{-16}$) and for 0 d versus 7 d (P < $2.2 \times 10^{-16}$). **(D)** Top: RNAPII activity observed by immunostaining in the RPE-1 cells without (untreated control), with serum starvation for 3 d, or with the re-addition of serum. Bottom: Quantification of RNAPII Ser5P signal intensity is shown as box plots. The median intensities of Ser5P are 54.5 ($n$ = 101) in

untreated control, 38.0 ($n$ = 103) in 3-d starvation, and 56.3 ($n$ = 79) in re-addition. Note that RNAPII activity decreased in the G0 phase and was restored with serum re-addition. The Wilcoxon rank sum test shows N.S. (P = 0.20), and ***, P < 0.0001 for untreated versus starvation (P < 2.2 × 10$^{-16}$) and for starvation versus serum re-addition (P < 2.2 × 10$^{-16}$). **(E)** MSD plots (±SD among cells) of nucleosomes in RPE-1 cells without (black) or with serum starvation for 3 d (light blue), and 1 d after serum re-addition (orange). For each condition, $n$ = 39–40 cells. The up-regulated chromatin dynamics were restored to the untreated level upon serum re-addition. The Kolmogorov–Smirnov test shows N.S. (P = 0.93), and ***, P < 0.0001 for untreated versus 3-d starvation (P = 3.2 × 10$^{-7}$) and for 3-d starvation versus serum re-addition (P = 4.8 × 10$^{-6}$).

constrained by the hubs and glues (Fig. 9 C, arrow), lowered the ensemble average of total chromatin dynamics (Fig. 9 B). Such a prediction is consistent with our observations of global changes in chromatin dynamics due to the inhibition or elimination of specific transcriptional proteins.

Second, available data suggest that RNAPII-Ser5P is involved in connecting chromatin domains but not in chromatin domain formation (Fig. 8 A, center). Our previous super-resolution imaging work showed that transcription inhibition did not alter the chromatin domain (or nucleosome cluster) organization while the domain formation was governed by both local nucleosome–nucleosome interactions and nucleosome fiber looping through cohesin (Fig. 8 A, round inset; Nozaki et al., 2017). Consistent with

this, a recent study by Hi-C showed that transcription inhibition had no appreciable effect on chromatin loop domain formation (Vian et al., 2018a). Furthermore, RNAPII-Ser5P clusters are often localized outside of the chromatin domains (Markaki et al., 2010; Nozaki et al., 2017; Xu et al., 2018). Since nucleosomes within the domains seem to move coherently in living cells (Nozaki et al., 2017), RNAPII-Ser5P would constrain movements of nucleosomes in the domain together, leading to the global reduction of chromatin motion (Fig. 8 A, center). Indeed, recently a spatial correlation of chromatin movements upon transcription was reported (Zidovska et al., 2013; Shaban et al., 2018).

Finally, our model is compatible with the classical transcription factory model (Buckley and Lis, 2014; Feuerborn and

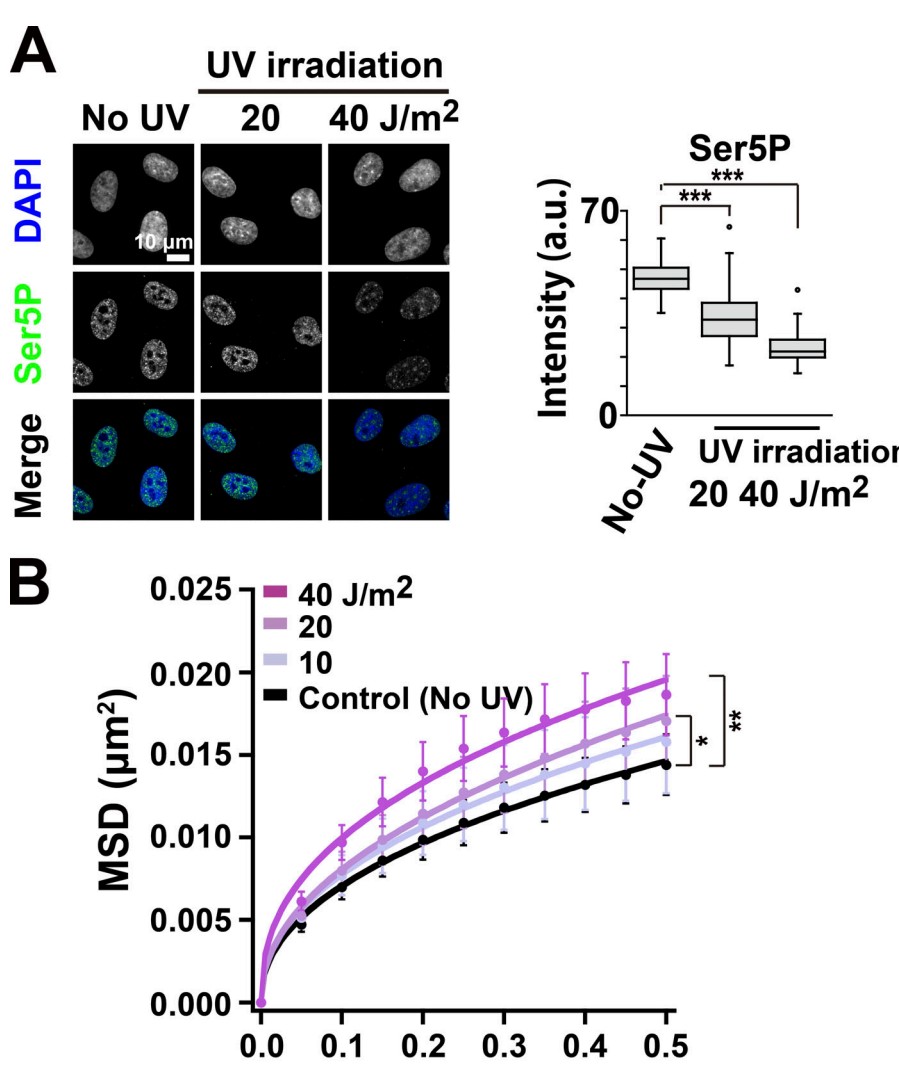

Figure 7. **UV-induced increase in chromatin dynamics. (A)** Left: RNAPII activity of RPE-1 cells before (no UV) and after 20-or 40-J/m$^2$ UV irradiation observed by immunostaining of Ser5P in RNAPII. Right: Quantifications of RNAPII Ser5P signal intensity are shown as box plots. The median intensities of Ser5P are 46.7 ($n$ = 114) in control, 32.7 ($n$ = 94) in 20 J/m$^2$, and 21.8 ($n$ = 89) in 40 J/m$^2$. Note that RNAPII activity decreased after the UV irradiation. ***, P < 0.0001 by the Wilcoxon rank sum test for no UV versus 20 J/m$^2$ (P < 2.2 × 10$^{-16}$) and for no UV versus 40 J/m$^2$ (P < 2.2 × 10$^{-16}$). **(B)** MSD plots (±SD among cells) of nucleosomes in RPE-1 cells before (no UV, black), after 10-, 20-, and 40-J/m$^2$ UV irradiation (from light to dark purples). $n$ = 12 cells in 10 J/m$^2$; $n$ = 10 cells in 20 J/m$^2$; $n$ = 9 cells in 40 J/m$^2$; $n$ = 27 cells in no UV. Note that the chromatin dynamics increased 6 h after UV irradiation. The Kolmogorov–Smirnov test shows **, P < 0.001 for untreated control versus 40 J/m$^2$ UV (P = 1.9 × 10$^{-4}$) and *, P < 0.05 for untreated control versus 20 J/m$^2$ UV (P = 0.028).

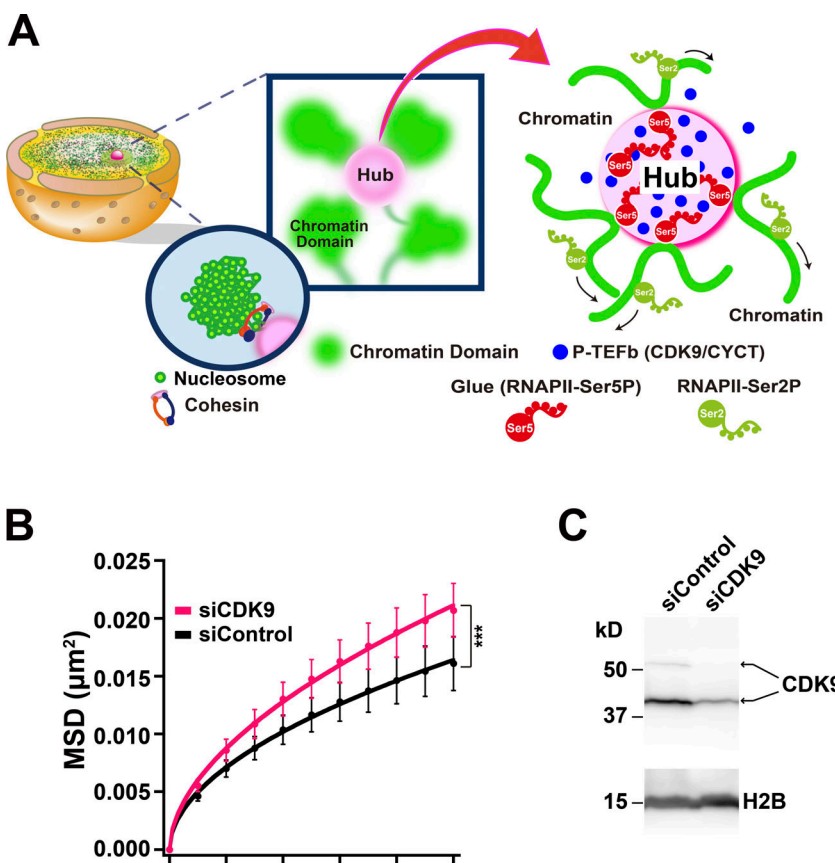

Figure 8. **A model for chromatin networking via RNAPII-Ser5P. (A)** A model for the formation of a loose spatial genome chromatin network via RNAPII-Ser5P, which can globally constrain chromatin dynamics. The P-TEFb complex (blue sphere in right panel) consisting of CYCT and CDK9 kinase, which interacts with RNAPII, forms a number of dynamic clusters/droplets in living cells (pink spheres in the center and right panels; Ghamari et al., 2013). Center: The P-TEFb cluster (pink sphere) can work as a hub to weakly connect multiple chromatin domains (green spheres) for a loose spatial genome network. Right: RNAPII-Ser5P (red) can function in the hub as glue for the weak interactions between P-TEFb (blue spheres) and transcribed DNA regions (green lines; Ghamari et al., 2013). Because after phosphorylation of RNAPII by P-TEFb, RNAPII-Ser2P seems to leave the hubs (P-TEFb clusters) for the elongation and processing process (Ghamari et al., 2013), it is unlikely to function as the glue for the connections (right). Note that this scheme is highly simplified. Besides the P-TEFb clusters, other clusters, including transcription factors, Mediator, and active RNAPII (Boehning et al., 2018; Boija et al., 2018; Cho et al., 2018; Chong et al., 2018; Sabari et al., 2018), might also work as hubs. **(B)** MSD plots (±SD among cells) of nucleosomes in CDK9-KD RPE-1 cells (siCDK9, pink) and control (siControl, black). For each condition, $n = 20$ cells. Note that the KD of CDK9 increased the chromatin dynamics. ***, $P < 0.0001$ ($P = 1.3 \times 10^{-6}$) by the Kolmogorov–Smirnov test. **(C)** CDK9 reduction in RPE-1 cells after RNA interference was verified by immunoblotting. H2B protein was used as a loading control.

Cook, 2015) and with recent studies showing that RNAPII and other factors form dynamic clusters/droplets, presumably by a phase separation process (Boehning et al., 2018; Boija et al., 2018; Cho et al., 2018; Chong et al., 2018; Lu et al., 2018; Sabari et al., 2018). Within the clusters/droplets, RNAPII and other transcription factors might be concentrated together to promote functional interactions between them, leading to highly efficient phosphorylation of the CTD and subsequent entry into elongation (Fig. 8 A; Cho et al., 2018; Chong et al., 2018; Lu et al., 2018; Sabari et al., 2018). In this context, the hub of the clusters/droplets may mediate chromatin domain contacts (Fig. 8 A) and further intrachromosomal and interchromosomal interactions for global control of gene transcription. Our model also predicts that transcriptional regulatory elements bound to the hubs and RNAPII-Ser5P would exit the hubs during the transition to transcription elongation. In such a case, the movements of these regulatory elements should markedly increase. Notably, recent elegant fluorescent labeling of transcriptional regulatory elements or neighboring regions revealed that their movements indeed increased upon transcription activation (Gu et al., 2018).

While the present study revealed the existence of the loose spatial genome network glued by active RNAPII (Fig. 8 A), we inferred that the up-regulation of the local chromatin fluctuation in the transcriptionally suppressive cells has a physiological relevance. Since our previous Monte Carlo simulation study suggested that an increase in local chromatin dynamics can

facilitate chromatin accessibility of transcription factors and other proteins (Hihara et al., 2012), in the transcriptionally suppressive cells or resting G0 cells the chromatin becomes more dynamic and may be in a high competency state for rapid and efficient recruitment of transcription factors to turn on certain genes in response to extracellular signals. A similar mechanism might be at work in UV-irradiated cells to recruit DNA repair machinery to damage sites.

## Materials and methods

### Plasmid construction

pPB-CAG-H2B-PA-mCherry plasmid was constructed on the basis of pPB-CAG-IB (provided from Sanger Institute with a materials transfer agreement) and pEF-1α-H2B-PA-mCherry (Nozaki et al., 2017). From pEF-1α-H2B-PA-mCherry, the coding region of H2B-PA-mCherry was amplified with the addition of the XhoI site to both ends using the following primer pairs: 5′-CCGCTCGAGATG CCAGAGCCAGCGAAGTC-3′ and 5′-CCGCTCGAGTTACTTGTACAG CTCGTCCATGCCG-3′.

The amplified H2B-PA-mCherry was inserted into pPB-CAG-IB digested with XhoI.

To construct pPB-CAG-H2B-HaloTag plasmid, the coding region of PA-mCherry in pPB-CAG-H2B-PA-mCherry plasmid was replaced with HaloTag. The HaloTag fragment was amplified from pFC14A-HaloTag CMV Flexi vector (G965A; Promega)

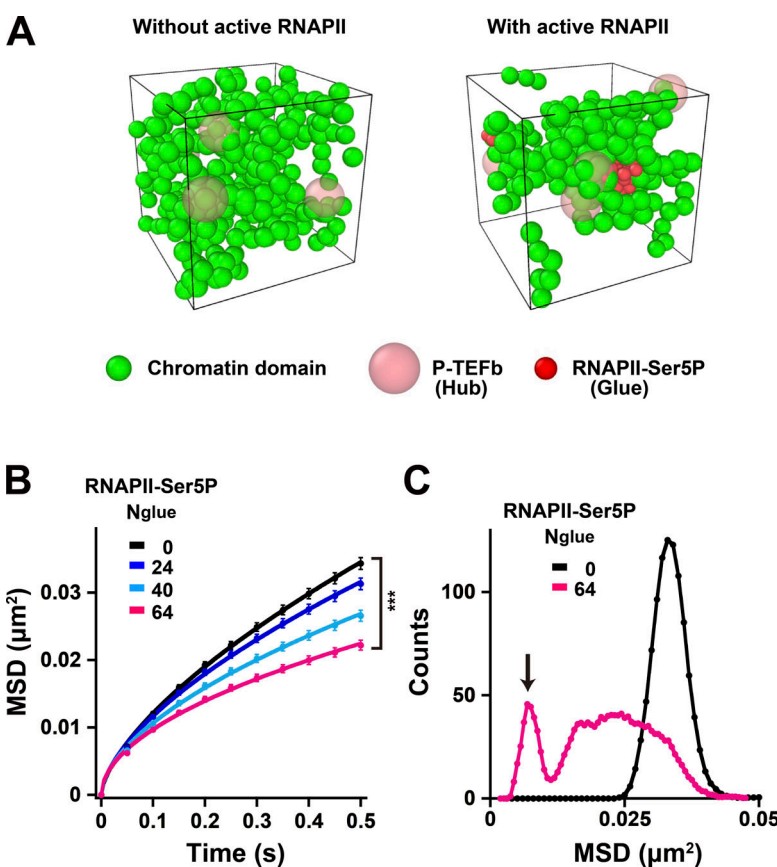

**A**

Without active RNAPII          With active RNAPII

● **Chromatin domain**     ● **P-TEFb (Hub)**     ● **RNAPII-Ser5P (Glue)**

**B**

RNAPII-Ser5P

Nglue
- ■ 0
- ■ 24
- ■ 40
- ■ 64

MSD ($\mu m^2$) vs Time (s)

***

**C**

RNAPII-Ser5P

Nglue
- ■ 0
- ■ 64

Counts vs MSD ($\mu m^2$)

Figure 9. **Computational modeling of chromatin domain network via active RNAPII.** Brownian dynamics of four chains of chromatin domains (green spheres) connected by springs (invisible) and four hubs, clusters of P-TEFb (pink spheres), were simulated in a box with the side length of 1.5 µm with the periodic boundary condition. Chromatin chains bind $N_{glue}$ RNAPII-Ser5P (red spheres), and attractive interactions were assumed between RNAPII-Ser5P–bound chromatin and P-TEFb clusters. **(A)** Left: A snapshot of the Brownian dynamics with no glues (without RNAPII-Ser5P). Coarse-grained 100-kb chromatin domains (green spheres) connected by invisible springs and the hubs of P-TEFb clusters (faint pink spheres) are shown. See also Video 5. Right: A snapshot of the Brownian dynamics with 64 glues of RNAPII-Ser5P (red spheres). In each of the four chromatin chains, 80 beads are connected by springs with each bead representing a 100-kb chromatin domain. For better visualization of their spatial distributions, green, red, and pink spheres are drawn with the radius of 90, 60, and 180 nm, respectively, although the SDs of their density distributions assumed in the model are ≈ 70, 70, and 210 nm. See also Video 6. **(B)** MSD plots calculated with various glue numbers bound on chromatins: $N_{glue}$ = 0 (black), 24 (dark blue), 40 (light blue), and 64 (pink). MSD plots were averaged over all the chromatin domains in the system and over 10 systems having different positions of RNAPII-Ser5P–binding sites. The plots were fitted as a subdiffusive curve: MSD = $0.054t^{0.64}$ without glues; MSD = $0.031t^{0.50}$ with $N_{glue}$ = 64. The bars represent standard errors. ***, P < 0.0001 (P = $1.0 \times 10^{-8}$) by Student's $t$ test. **(C)** The MSD (0.5 s) distribution plots of total beads with (pink) and without RNAPII-Ser5P glues (black). Note that there are a slow and fast bead fraction with RNAPII-Ser5P glues. The slow fraction (arrow) lowers the ensemble average of total beads dynamics in the system. Images in A were drawn with the software OVITO (Stukowski, 2010).

with the addition of BamHI and HpaI sites to the ends in three steps by using the following primer pairs: 5′-TGGAGGTGGAGG CTCTGGTGGCGGTTCCGAAATCGGTACTGG-3′ and 5′-CCAGTT AACTTAACCGGAAATCTCCAGAG-3′; 5′-TGGTTCAGGCGGAGG TGGAAGTGGAGGTGGAGGCTCTGGTGG-3′ and 5′-CCAGTTAAC TTAACCGGAAATCTCCAGAG-3′; and 5′-CGCGGATCCATCTGG TGGAGGTGGTTCAGGCGGAGGTGGAAG-3′ and 5′-CCAGTTAAC TTAACCGGAAATCTCCAGAG-3′.

pPB-CAG-H2B-PA-mCherry plasmid and the amplified Hal-oTag fragment were digested with BamHI and HpaI and ligated.

Construction of EF-1α-Venus-Nup107-cHS4-FRT was described previously (Maeshima et al., 2010b). In brief, to first replace the CMV promoter of the pcDNA5/FRT vector (V601020; Thermo Fisher Scientific) with the EF-1α promoter, the EF-1α promoter region of the plasmid pEF5/FRT/V5-DEST (V602020; Thermo Fisher Scientific) was PCR-amplified and inserted into pcDNA5/FRT to make EF-1α-FRT. Then, the coding region of the human Nup107 cDNA (Dr. V. Doye, Curie Institute, Paris, France) was PCR-amplified and cloned into the pVenus-C3 plasmid (Nagai et al., 2002; Dr. T. Nagai, Osaka University, Osaka, Japan) to make the Venus–Nup107 construct, which was excised and ligated into pEF-1α-FRT to create the vector pEF-1α-Venus-Nup107-FRT. To obtain stable expression of the inserted cDNA, we inserted the chicken insulator fragment cHS4 (Dr. G. Felsenfeld, National Institutes of Health, Bethesda, MD) downstream of Venus–Nup107.

### Cell lines

RPE-1 (CRL-4000; ATCC) and HeLaS3 cells (Maeshima et al., 2006) were cultured at 37°C in 5% $CO_2$ in DMEM (D5796-500ML; Sigma-Aldrich) supplemented with 10% FBS (FB-1061/500; Biosera; Hihara et al., 2012; Nozaki et al., 2017). DLD-1 cells (CCL-221; ATCC) and HCT116 cells (CCL-247; ATCC) were cultured at 37°C in 5% $CO_2$ in RPMI-1640 medium (R8758-500ML; Sigma-Aldrich) supplemented with 10% FBS and McCoy's 5A medium (SH30200.01; HyClone) supplemented with 10% FBS, respectively.

To establish RPE-1 cell lines stably expressing H2B-Halo or H2B-PA-mCherry, the transposon system was used. The constructed plasmid pPB-CAG-IB-H2B-HaloTag or pPB-PGKneo-EF1-H2B-PA-mCherry (Nozaki et al., 2017) was cotransfected with pCMV-hyPBase (provided from Sanger Institute with a materials transfer agreement) to RPE-1 cells with the Effectene transfection reagent kit (301425; QIAGEN). Transfected cells were then selected with 10 µg/ml blasticidin S (029–18701; Wako; pPB-CAG-IB-H2B-HaloTag) or 600 µg/ml G418 (ALX-380-013-G001; ENZ; H2B-PA-mCherry).

To construct parental DLD-1 and HCT116 cells expressing Os-TIR1, a donor plasmid containing Tet-OsTIR1 (pMK243; Natsume et al., 2016) was integrated at the safe harbor AAVS1 locus by using CRISPR–Cas9-based gene editing (Fig. 5 B, top; Natsume et al., 2016; Yesbolatova et al., 2019 *Preprint*). Subsequently, as shown in the bottom panel of Fig. 5 B, we transfected a CRISPR-Cas9

plasmid targeting at the first methionine site of the *POLR2A* gene (CCT/CCGCCATGCACGGGGGTGGC) together with a donor harboring a Hygro-P2A-mAID-mClover cassette flanked with a 500-bp homology arm by using the FuGENE HD Transfection Reagent (E2311; Promega). After selection with 200 µg/ml of hygromycin (10687010; Thermo Fisher Scientific), colonies were isolated for further analysis by genomic PCR and Western blotting. The genomic PCR was performed using the following primers: 5′-CTTCTCTCCCTGTCACTTCAAAAGG-3′ and 5′-CTATTC GAGACAAATGCCATTAAC-3′.

The subsequent processes to introduce H2B-HaloTag were done as described above.

To obtain Nup107-Venus expressed HeLa cells, EF-1α-Venus-Nup107-cHS4-FRT was transfected into HeLa S3 cells (Maeshima et al., 2010b). After selection with 200 µg/ml hygromycin (10687010; Thermo Fisher Scientific), colonies were isolated for further analysis.

## Biochemical fractionation of nuclei from cells expressing H2B-HaloTag

Nuclei were isolated from RPE-1 cells expressing H2B-HaloTag as described previously (Maeshima et al., 2016; Nozaki et al., 2017). Briefly, collected cells were suspended in nuclei isolation buffer (3.75 mM Tris-HCl [pH 7.5], 20 mM KCl, 0.5 mM EDTA, 0.05 mM spermine, 0.125 mM spermidine, 1 µg/ml Aprotinin [T010A; TaKaRa], and 0.1 mM PMSF [P7626-1G; Sigma-Aldrich]) and centrifuged at 1,936 *g* for 7 min at room temperature. The cell pellets were resuspended in nuclei isolation buffer and again centrifuged at 1,936 *g* for 7 min at room temperature. The cell pellets were then resuspended in nuclei isolation buffer containing 0.025% Empigen (45165-50ML; Sigma-Aldrich; nuclei isolation buffer+) and homogenized immediately with 10 downward strokes by using a tight Dounce-pestle. The cell lysates were centrifuged at 4,336 *g* for 5 min. The nuclei pellets were washed in nuclei isolation buffer+. The nuclei were incubated on ice for 15 min in a series of buffers: HE (10 mM Hepes-NaOH [pH 7.5], 1 mM EDTA, and 0.1 mM PMSF), HE + 100 mM NaCl, HE + 500 mM NaCl, HE + 1 M NaCl, and HE + 2 MNaCl. After incubation with salt, centrifugation was performed to separate the nuclear solutions into supernatant and pellet fractions. The proteins in the supernatant fractions were precipitated by using 17% trichloroacetic acid (208-08081; Wako) and cold acetone. Both pellets were suspended in SDS-PAGE buffer and subjected to 12.5% SDS-PAGE and subsequent Coomassie brilliant blue (031-17922; Wako) staining and Western blotting by using rabbit anti–H2B (ab1790; Abcam) and rabbit anti–HaloTag (G9281; Promega) antibodies.

## HaloTag labeling
To confirm the expression, H2B-Halo molecules were fluorescently labeled with 100 nM HaloTag TMR Ligand (8251; Promega) overnight at 37°C in 5% $CO_2$. Subsequently, the cells were fixed with 1.85% FA (064–00406; Wako) for 15 min and then permeabilized with 0.5% Triton X-100 (T-9284; Sigma-Aldrich) for 15 min and stained with 0.5 µg/ml DAPI (10236276001; Roche) for 5 min before PPDI (20 mM Hepes [pH 7.4], 1 mM $MgCl_2$, 100 mM KCl, 78% glycerol, and 1 mg/ml paraphenylene diamine

[695106-1G; Sigma-Aldrich]) mounting. For single-molecule imaging, H2B-Halo molecules were fluorescently labeled with 80 pM HaloTag TMR ligand for 20 min at 37°C in 5% $CO_2$, washed with 1× HBSS (H1387; Sigma-Aldrich) three times, and then incubated in medium without phenol red for more than 30 min before live-cell imaging.

## Single-nucleosome imaging microscopy
Established cell lines were cultured on poly-L-lysine (P1524-500MG; Sigma-Aldrich) coated glass-based dishes (3970-035; Iwaki). H2B-Halo molecules were fluorescently labeled with 80 pM HaloTag TMR ligand (8251; Promega) as described above. RPE-1, DLD-1, and HCT116 cells were observed in the DMEM (21063-029; Thermo Fisher Scientific), RPMI-1640 (11835-030; Thermo Fisher Scientific), and McCoy's 5A (1-18F23-1; Bio-Concept), respectively. All of the mediums were phenol red free and supplemented with 10% FBS. For serum starvation, cells were cultured in DMEM supplemented with 1% BSA (A9647; Sigma-Aldrich) for 3–7 d.

To maintain cell culture conditions (37°C, 5% $CO_2$, and humidity) under the microscope, a live-cell chamber INU-TIZ-F1 (Tokai Hit) and GM-8000 digital gas mixer (Tokai Hit) were used. Single nucleosomes were observed by using an inverted Nikon Eclipse Ti microscope with a 100-mW Sapphire 561-nm laser (Coherent) and sCMOS ORCA-Flash 4.0 camera (Hamamatsu Photonics). Fluorescently labeled H2B-Halo(TMR) in living cells were excited by the 561-nm laser through an objective lens (100× PlanApo TIRF, NA 1.49; Nikon) and detected at 575–710 nm. An oblique illumination system with the TIRF unit (Nikon) was used to excite H2B-Halo-TMR molecules within a limited thin area in the cell nucleus and reduce a background noise. Sequential image frames were acquired using MetaMorph software (Molecular Devices) at a frame rate of 50 ms under continuous illumination.

## Single-nucleosome tracking analysis
The methods for image processing, single-molecule tracking, and single-nucleosome movement analysis were described previously (Nozaki et al., 2017). Sequential images were converted to 8-bit grayscale, and the background noise signals were subtracted with the rolling ball background subtraction (50) of Fiji software (Schindelin et al., 2012). The nuclear regions in the images were manually extracted. Following this step, the centroid of each fluorescent dot in each image was determined, and its trajectory was tracked with u-track (MATLAB package; Jaqaman et al., 2008). To generate photoactivated localization microscopy (PALM) images, the individual nucleosome positions were mapped using R software (65 nm/pixel) on the basis of the u-track data. For single-nucleosome movement analysis, the displacement and MSD of the fluorescent dots were calculated on the basis of their trajectory using a Python program (the script is available in the Online supplemental material). The originally calculated MSD was in 2D. To obtain the 3D value, the 2D value was multiplied by 1.5 (4 to 6 Dt). To make the heat map of chromatin dynamics, the median nucleosome movements (during 50 ms) in 3 × 3 pixels (65 nm/pixel) were plotted with a color scale from blue to red by using R (the script is available in the

Online supplemental material). Statistical analyses of the obtained single-nucleosome MSD between various transcription conditions were performed using R.

## Chemical treatment

For transcription inhibition, cells were treated with transcription inhibitors, 100 µM or 20 µM DRB (D1916-10MG; Sigma-Aldrich) for 2 h, 100 µg/ml α-AM (A2263-1MG; Sigma-Aldrich) for 2 h or 20 µg/ml for 6 h (only for EU labeling) or 2 h, or ActD (A9415-2MG; Sigma-Aldrich) 0.5 µg/ml or 0.01 µg/ml for 2 h. To inhibit RNA polymerase I, cells were treated with 1 µM CX5461 (M66052-2s; Xcess Biosciences Inc.) for 2 h. To inhibit splicing, cells were treated with 30 ng/ml Pladienolide B (sc-391691; Santa Cruz) for 2 h. To degrade mAID-mClover-RPB1 rapidly, cells were incubated in medium supplemented with 1 µg/ml doxycycline (631311; BD) for 23 h and then treated with 500 µM indole-3-acetic acid (19119-61; Nacalai), a natural auxin, in the presence of the doxycycline for 1 h (DLD-1) or 2 h (HCT116). After the treatment, cells were imaged or chemically fixed.

## Immunoblotting

Cells were lysed in Laemmli sample buffer (Laemmli, 1970) supplemented with 10% 2-mercaptoethanol (133-1457; Wako) and incubated at 95°C for 5 min to denature proteins. Then, SDS-PAGE was performed to separate the proteins. Proteins in the gel were transferred to an Immobilon-P membrane (IPVH00010; Merck) and blocked with PBS-T containing 5% nonfat milk (190-12865; Wako) for 30 min at room temperature. Subsequently, the proteins were blotted with antibodies at the indicated dilutions: rabbit anti–histone H2B (ab1790; Abcam) at 1:10,000 or 1:20,000; rabbit anti–HaloTag (G9281; Promega) at 1:1,000; mouse anti–CTD of RPB1 (ab817; Abcam) at 1:1,000; mouse anti–α-Tubuline (T6199; Sigma-Aldrich) at 1:5,000; mouse anti–CDK9 (sc-13130; Santa Cruz) at 1:500; horseradish peroxidase-linked goat anti–rabbit IgG whole antibody (170-6515; Bio-Rad) at 1:5,000 for anti-H2B and anti-HaloTag; HRP-linked goat anti–mouse IgG whole antibody (170-6516; Bio-Rad) at 1:5,000 for anti-CTD, anti-α-Tubuline, and anti-CDK9. Signal detection was performed by using the Immobilon Western Chemiluminescent HRP substrates (WBKLS0500; Merck) with a chemiluminescence CCD imaging system EZ-Capture MG (ATTO).

## Indirect immunofluorescence

Immunostaining was performed as described previously (Hihara et al., 2012), and all processes were performed at room temperature. Cells were fixed in 1.85% FA in PBS for 15 min and then treated with 50 mM glycine in HMK (20 mM Hepes [pH 7.5] with 1 mM $MgCl_2$ and 100 mM KCl) for 5 min and permeabilized with 0.5% Triton X-100 in HMK for 5 min. After washing twice with HMK for 5 min, the cells were incubated with 10% normal goat serum (NGS; 143-06561; Wako) in HMK for 30 min. The cells were incubated with diluted primary antibodies: mouse anti–Ki67 (1:1,000, NA59; Oncogene), mouse anti–Topoisomerase IIα (1:1,000, M042-3; MBL), mouse anti–phosphorylated Ser5 of RNAPII (1:1,000, RNAPII-Ser5P provided by Dr. H. Kimura; clone CMA603 described in Stasevich et al., 2014), rabbit anti–RNAPII-Ser5P (1:2,000, ab5131; Abcam), and mouse anti–RNAPII-Ser2P

(1:1,000 or 1:2,000, Dr. H. Kimura; clone CMA602 described in Stasevich et al., 2014) in 1% NGS in HMK for 1 h. After being washed with HMK four times, the cells were incubated with diluted secondary antibodies: goat anti–mouse IgG Alexa Fluor 488 (1:1,000, A11029; Thermo Fisher Scientific), goat anti–mouse IgG Alexa Fluor 594 (1:500 or 1:1,000, A11032; Thermo Fisher Scientific), goat anti–rabbit IgG Alexa Fluor 594 (1:1,000, A11037; Thermo Fisher Scientific), and goat anti–rabbit IgG Alexa Fluor 647 (1:1,000, A21245; Thermo Fisher Scientific) in 1% NGS in HMK for 1 h followed by a wash with HMK four times. For DNA staining in fixed cells, 0.5 µg/ml DAPI was added to the cells for 5 min followed by washing with HMK. The stained cells were mounted in PPDI and sealed with a nail polish (T and B; Shiseido). Most of the images were obtained using a DeltaVision Elite microscopy imaging system (Applied Precision) described below, and some (Fig. S4, A and B) were acquired at room temperature with Nikon Eclipse Ti microscope equipped with Olympus UPLAMSAPO 60×W (NA 1.20) and sCMOS ORCA-Flash 4.0 camera (Hamamatsu Photonics) using NIS Elements BR 4.20 (Nikon) software.

## Imaging and quantification of immunostaining images

Image z-stacks (every 0.2 µm in the z direction, 20–25 sections in total) of the immunostained samples were obtained at room temperature by using the DeltaVision Elite microscopy with Olympus PlanApoN 60× objective (NA 1.42) and sCMOS camera. InsightSSI light (~50 mW) and the four-color standard filter set were equipped. Acquisition software was Softworx in the DeltaVision. Because the signals were not distributed homogeneously across z-stacks, some images in the stacks were projected to cover whole nucleus (usually seven images) using Softworx and used as a source images. Nucleoplasm regions were extracted on the basis of the DNA (DAPI) staining regions. For active RNAPII staining, the mean intensities of the nuclear signals after background subtraction (the nuclear signals without primary antibody treatment or the signals outside nuclei) were calculated and plotted. For immunostaining with Ki67 and Topo II, the numbers of nuclei with intensity higher than a threshold value were counted and plotted.

## EU labeling and quantification

EU incorporation was performed by using Click-iT RNA Alexa Fluor 594 Imaging Kit (C10330; Thermo Fisher Scientific) according to the manufacturer's instructions. Cells were incubated with 500 µM EU for 1 h during the period of chemical treatment. The cells were then fixed with 3.7% FA for 15 min and permeabilized with 0.5% Triton X-100 for 15 min. Incorporated EU was labeled by Click-iT reaction according to the manufacturer's instructions. The cells were stained with 0.5 µg/ml DAPI for 5 min and mounted in PPDI. Image stacks were obtained by using the DeltaVision microscopy (Applied Precision) as described above and projected to cover the whole nucleus (seven images). Nucleoplasm regions were extracted on the basis of DNA (DAPI) staining. The median of each cell's mean intensity of the extracted nuclear signals after background subtraction (the signals outside nuclei) were calculated and plotted.

## RNA preparation and quantitative real-time PCR

Sample RNA purification was performed by using the Click-iT Nascent RNA Capture Kit (C10365; Thermo Fisher Scientific). To label the nascent RNA with 5-EU, cells cultured on six wells were incubated with DMEM supplemented with 500 µM of 5-EU for 1 h at 37°C under 5% $CO_2$. After labeling, RNA was extracted from the cells by using TRIzol (10296-010; Thermo Fisher Scientific), and subsequently, labeled RNA was biotinylated by the click reaction. The biotinylated RNA was purified by streptavidin beads. To prepare cDNA, the purified RNA was reverse transcribed using SuperScript III First-Strand Synthesis SuperMix (18080-400; Thermo Fisher Scientific), then the *CDK6* gene region was amplified from the purified cDNA by using TB Green Premix Ex Taq (RR820A; Takara) and, simultaneously, SYBR Green signal was detected by using Thermal Cycler Dice Real Time System TP800 (Takara). 18S rDNA was used as a reference gene. Primers are as follow: h18S rDNA_Fw: 5′-GTTGGTGGA GCGATTTGTCTGGTT-3′; h18S rDNA_Rv: 5′-TATTGCTCAATC TCGGGTGGCTGA-3′; hCDK6_Int2_Fw: 5′-TGCAGCTGTGCAACT TAGA-3′; hCDK6_Int2_Rv: 5′-GTTGGCTTATCCTGTCCCTAAA-3′; hCDK6_Ex2-3 spliced_Fw: 5′-ATGCCGCTCTCCACCAT-3′; hCDK6_Ex2-3 spliced_Rv: 5′-ACATCAAACAACCTGAC-3′.

## OI-DIC imaging and total density estimation in the nucleoplasm of RPE-1 cells

We estimated intracellular density distribution from obtained optical path difference (OPD) maps using OI-DIC microscopy as described previously (Imai et al., 2017). A simple scheme is shown in Fig. S2 E. In brief, the following two steps were performed. First, we calculated the refractive index (RI) from the OPD. Because the OPD is proportional to the thickness of a sample and the difference in RI between the sample and the surrounding solution (Fig. S2 E, left), we calculated the RI of samples on the basis of the RI of the surrounding solution and sample thickness (Fig. S2 E, right). Second, we obtained the density of the sample from its RI because the RI of a sample is proportional to its density. The dry mass density in live cells, which consists mainly of proteins and nucleic acids, was calculated from their RI using a single calibration curve (Fig. S3 in Imai et al., 2017). To estimate the densities of the total cell contents, we measured the average thickness of the cytoplasm stained by Calcein-AM (C396; Dojindo) and the nucleus stained by Hoechst 33342 (H342; Dojindo) in the live cells by FLUOVIEW FV1000 confocal laser scanning microscope (OLYMPUS) equipped with Olympus UPLAMSAPO 60×W objective (NA 1.20) at room temperature. Hoechst 33342 and Calcein-AM fluorescence signals were acquired as 3D image stacks (500 nm × 32 sections). The thicknesses of three regions (nucleus, cytoplasm, entire cell; see supplemental Fig. S1 in Imai et al., 2017) were measured in each cell from the acquired stack images by Fiji software. To obtain the RI of cytoplasm ($RI_{cy}$), we used the RI of the surrounding culture medium ($RI_m$, 1.3375). For the RI of the nucleus, we used our calculated values of $RI_{cy}$. These estimates were created using Fiji software (Schindelin et al., 2012).

## Measurements of free $Mg^{2+}$ in the RPE-1 cells

Magnesium Green-AM (M3735; Thermo Fisher Scientific) was applied to the DMEM culture medium at 10 µg/ml with 0.02%

Pluronic F-127 (P3000MP; Thermo Fisher Scientific), and the cells were incubated at 37°C for 1 h. The cells were then washed twice with HBSS (14170112; Thermo Fisher Scientific) and further incubated in fresh HBSS at 37°C for 15 min to complete hydrolysis of the acetoxymethyl ester form. Magnesium Green fluorescence was measured by DeltaVision equipped with an Olympus PlanApoN 60× objective (NA 1.42) and an sCMOS camera with an FITC filter by using Softworx software. The cell culture conditions (37°C, 5% $CO_2$, and humidity) were maintained in a live-cell chamber under the microscope. The nucleoplasm intensity was measured by using Softworx software after subtraction of background signals outside cells and plotted.

## UV irradiation

Cells grown to ∼40–60% confluency were exposed to ultraviolet C irradiation in 200 µl medium without phenol red by using CX-2000 (Fisher Scientific). After irradiation, the cells were cultured for 6 h before imaging or fixation.

## RNA interference

Transfection of siRNA was performed using Lipofectamine RNAiMAX (13778-075; Thermo Fisher Scientific) according to the manufacturer's instructions. The medium was changed to a fresh medium 16 h after transfection. The transfected cells were used for subsequent studies 48 h after transfection. The siRNA oligonucleotide targeting CDK9 sequence (s2834; Thermo Fisher Scientific; Sense: 5′-UGAGAUUUGUCGAACCAAATT-3′) was used. As a control, an oligonucleotide (4390843; Thermo Fisher Scientific; the sequence is undisclosed) was used.

## Computational modeling

Dynamics of four chromatin chains were simulated in a cubic box with side length of 1.5 µm by applying the periodic boundary condition. Each chain is a coarse-grained bead-and-spring chain composed of 80 beads, which are connected by springs whose energy is defined by Eq. 1:

$$u_{\text{spring}}(r_{ii+1}) = \frac{K}{2}(r_{ii+1} - \sigma)^2, \tag{1}$$

where $r_{ii+1}$ is the distance between centers of $i$th and $i$+first beads with $\sigma$ = 150 nm. $K = 90k_B T_0/\sigma^2$ with $T_0$ = 37°C and $k_B$ is the Boltzmann constant. Each bead represents a 100-kb chromatin domain so that the simulated density of chromatins in the box is similar to that of human genome in a sphere of 10-µm diameter. Chromatin in each bead was assumed to spread with a Gaussian distribution of the SD $\sigma_b \approx 70$ nm, so as to make its radius of gyration $R_g = \sqrt{3\sigma_b} \approx 120$ nm similar to the observed radius $R_g \approx$ 100–200 nm in a microscopic measurement (Boettiger et al., 2016). Four hubs (P-TEFb clusters) were dispersed in the box. We assumed that the density distribution of P-TEFb factors in each cluster is represented by a Gaussian of the SD $\sigma_c$ with $2\sigma_c \approx 420$ nm, which should correspond to the microscopically observed diameter of 300–500 nm of the RNAPII-Ser5P assembly (Ghamari et al., 2013). Since transcription starting sites are often found at boundaries of topological domains of several 100-kb size (Dixon et al., 2012), we assumed that the system has 64 sites that can bind RNAPII. We randomly selected 64 chromatin

beads from 80 × 4 = 320 beads and assumed that $N_{glue}$ among 64 sites bind RNAPII-Ser5P. 10 different systems were generated by selecting 10 different sets of $N_{glue}$ locations on chromatin chains. To highlight the effects of gluing of chains to the hubs P-TEFb clusters, we did not consider the chain looping due to the cohesin binding or compartment formation in the present simplified model.

Repulsive interaction was assumed between chromatin beads, whose strength is proportional to the overlap of two density distributions of chromatins. Because the overlap of two Gaussians with the dispersion $\sigma_b{}^2$ is a Gaussian with the dispersion $\sigma_{bb}{}^2 = 2\sigma_b{}^2$, potential of the repulsive interaction $u_{bb}(r_{ij})$, between $i$ and $j$th beads should be a Gaussian with the SD $\sigma_{bb} = \sqrt{2\sigma_b{}^2} = 100$ nm. In the present modeling, the Gaussian function was approximated by a computationally more economical function as shown in Eq. 2:

$$u_{bb}(r_{ij}) = \begin{cases} \varepsilon_{bb}\left[1 - \left(\dfrac{r_{ij}}{3\sigma_{bb}}\right)^2\right]^3, & \text{for } r_{ij} < 3\sigma_{bb}, \\ 0, & \text{for } r_{ij} \geq 3\sigma_{bb}, \end{cases} \quad (2)$$

with the interaction strength $\varepsilon_{bb} = 3k_BT_0$. The repulsive interaction, $u_{cc}(r_{\mu\nu})$, between $\mu$ and $\nu$th clusters was defined to prevent them from collapsing together as shown in Eq. 3:

$$u_{cc}(r_{\mu\nu}) = \begin{cases} \varepsilon_{cc}\left[1 - \left(\dfrac{r_{\mu\nu}}{3\sigma_{cc}}\right)^2\right]^3, & \text{for } r_{\mu\nu} < 3\sigma_{cc}, \\ 0, & \text{for } r_{\mu\nu} \geq 3\sigma_{cc}, \end{cases} \quad (3)$$

with $\sigma_{cc} = \sqrt{2\sigma_c{}^2} = 300$ nm and $\varepsilon_{cc} = 40k_BT_0$. The attractive interaction, $u_{cb}(r_{\mu s})$, between the $\mu$th cluster and $s$th RNAPII-Ser5P-binding site with $1 \leq s \leq N_{glue}$ glues the chromatin chain to the P-TEFb cluster as shown in Eq. 4:

$$u_{cb}(r_{\mu s}) = \begin{cases} -\varepsilon_{cb}\left[1 - \left(\dfrac{r_{\mu s}}{3\sigma_{cb}}\right)^2\right]^3, & \text{for } r_{\mu s} < 3\sigma_{cb}, \\ 0, & \text{for } r_{\mu s} \geq 3\sigma_{cb}, \end{cases} \quad (4)$$

with $\sigma_{cb} = \sqrt{\sigma_c{}^2 + \sigma_b{}^2} = 225$ nm and $\varepsilon_{cb} = 20k_BT_0$.

Movement of chromatin beads and clusters was simulated by numerically integrating the Brownian dynamics (overdamped Langevin) equation at temperature $T_0 = 37°C$ with a discrete time step $10^{-4}$ s under the force derived from the potential as shown in Eq. 5:

$$U = \sum_i u_{spring}(r_{ii+1}) + \sum_{i \neq j} u_{bb}(r_{ij}) + \sum_{\mu \neq \nu} u_{cc}(r_{\mu\nu}) + \sum_{\mu=1}^{4}\sum_{s=1}^{N_{glue}} u_{cb}(r_{\mu s}), \quad (5)$$

With $0 \leq N_{glue} \leq 64$. The friction coefficients in the Brownian dynamics equation, $\gamma_b$ for a chromatin bead and $\gamma_c$ for a P-TEFb cluster, were determined by the Stokes-Einstein relation, $\gamma_b = 6\pi\sigma_b\eta$ and $\gamma_c = 6\pi\sigma_c\eta$, with viscosity of liquid water at 37°C, $\eta = 6.9 \times 10^{-4} \text{Pa} \cdot \text{s}$. At the initial time step, four chromatin chains were placed on the box walls, and four clusters were randomly positioned in the box. Then, the chains and clusters were moved at a high temperature, $T = 10T_0$, and the temperature was linearly decreased to $T = T_0$ by taking $1.5 \times 10^6$ steps. Then, the system was equilibrated at $T = T_0$ for another $1.5 \times 10^6$ steps, and after that, the data were sampled for the statistical evaluation. Results of 10 different systems defined by 10 randomly selected sets of $N_{glue}$ sites on chromatin chains with different initial positions of P-TEFb clusters were averaged to obtain the results shown in Fig. 9 B.

### Online supplemental material

Fig. S1 shows validations of H2B-Halo–labeled single nucleosomes, their position determination accuracy, and visualization of their dynamics. Fig. S2 shows RNAPII-Ser2P intensity quantification, single-nucleosome MSD with transcription inhibitor treatments, total material density imaging, free $Mg^{2+}$ imaging, and nuclear surface imaging. Fig. S3 shows rapid degradation of RNAPII in HCT116 cells and increased the chromatin dynamics. Fig. S4 shows immunostaining of Ki67, TopoIIα, and RNAPII-Ser2P and nucleosome dynamics in serum-starved cells. Fig. S5 shows immunostaining of Ki67 and RNAPII-Ser2P in serum-restored cells and RNAPII-Ser2P immunostaining in UV-irradiated cells. Video 1 shows data of single nucleosomes labeled with TMR in a living RPE-1 cell. Video 2 is a tracking example of the data in Video 1. Video 3 shows data of single nucleosomes labeled with TMR in the living RPE-1 cell treated with α-AM at 100 µg/ml for 2 h. Video 4 shows data of single nucleosomes labeled with TMR in the nuclear periphery of a living RPE-1 cell. Video 5 shows computational modeling for 1.5-s movement of four chromatin chains with domains and four hubs in the absence of the glues. Video 6 shows computational modeling for 1.5-s movement of four chromatin chains with domains, four hubs, and 64 glues. Two ZIP files containing some scripts are also provided online. For single-nucleosome movement analysis, the displacement and MSD of the fluorescent dots were calculated on the basis of its trajectory using a Python program. To make the heat map of chromatin dynamics, the median nucleosome movements (during 50 ms) in 3 × 3 pixels (65 nm/pixel) were plotted with a color scale from blue to red using R.

## Acknowledgments

We thank Dr. Y. Hiromi, Dr. S. Hirose, Dr. H. Seino, and Dr. S. Ide for critical reading of this manuscript. We thank Dr. S. Ide, Dr. D. Kaida, Dr. T. Nagai, Dr. V. Doye, Dr. G. Felsenfeld, and Dr. K. Horie for valuable help and materials. We also thank the Maeshima laboratory members for helpful discussions and support.

R. Imai and T. Nozaki are Japan Society for the Promotion of Science Fellows. R. Nagashima was supported by 2017 SO-KENDAI Short-Stay Study Abroad Program. This work was supported by a Japan Society for the Promotion of Science grant (16H04746), Takeda Science Foundation, RIKEN Pioneering Project, a Japan Science and Technology Agency Core Research for Evolutional Science and Technology grant (JPMJCR15G2), a

National Institute of General Medical Sciences grant (R01-GM101701), and National Institute of Genetics JOINT (2016-A2 (6)).

The authors declare no competing financial interests.

Author contributions: R. Nagashima, K. Hibino, and K. Maeshima designed the project; R. Nagashima performed most of the experiments on cell generation, imaging, and analyses; K. Hibino and M. Babokhov performed some experiments on cell generation, imaging, and analyses; S.S. Ashwin, S. Fujishiro, and M. Sasai performed computational modeling and analysis; T. Nozaki and R. Imai contributed to methodology establishment of single-nucleosome imaging and analysis; S. Tamura contributed to immunoblotting experiments and illustration; M. Shribak and T. Tani contributed to OI-DIC imaging; M.T. Kanemaki contributed to AID cell generation; H. Kimura contributed to some essential materials; R. Nagashima, K. Hibino, and K. Maeshima wrote the manuscript with input from all other authors.

Submitted: 18 November 2018

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
