## [Reviewer comments · The Journal of Cell Biology]

Single nucleosome imaging reveals loose genome chromatin networks via active RNA polymerase II

Ryosuke Nagashima, Kayo Hibino, S Ashwin, Michael Babokhov, Shin Fujishiro, Ryosuke Imai, Tadasu Nozaki, Sachiko Tamura, Tomomi Tani, Hiroshi Kimura, Michael Shribak, Masato Kanemaki, Masaki Sasai, and Kazuhiro Maeshima

Corresponding Author(s): Kazuhiro Maeshima, National Institute of Genetics

Review Timeline:

Submission Date:	2018-11-18
Editorial Decision:	2018-12-21
Revision Received:	2019-01-31
Editorial Decision:	2019-02-01
Revision Received:	2019-02-04

Monitoring Editor: Bas van Steensel

Scientific Editor: Melina Casadio

Transaction Report:

DOI: <https://doi.org/N/A>

December 21, 2018

Re: JCB manuscript #201811090

Dr. Kazuhiro Maeshima
National Institute of Genetics
Yata 1111
Mishima 411-8540
Japan

Dear Dr. Maeshima,

Thank you for submitting your manuscript entitled "Single nucleosome imaging reveals loose genome chromatin networks via active RNA polymerase II". Thank you very much for your patience with the review process - we sincerely apologize for the delay in communicating our decision to you. The manuscript was assessed by expert reviewers, whose comments are appended to this letter. We invite you to submit a revision if you can address the reviewers' key concerns, as outlined here.

You will see that the reviewers - and we agree - find the data important and interesting and support the work at the journal. They provided constructive comments that we have discussed editorially. We encourage you to tackle the reviewers' comments as follows:

- Rev#1 has helpful but minor points that should help improve the manuscript further. We do not think the rescue (wash out auxin) experiment is needed for publication in JCB, so data addressing this point experimentally will not be required for us to move forward with the paper at the journal - please address this point as you see fit. The possibility that UV causes DNA breaks that alter mobility is interesting, this should at least be discussed (it may be known how many DNA breaks a given UV dose causes).
- Rev#2's major points are constructive and could be addressed by rewriting the text, and/or by addressing these issues in the Discussion section. We do not think additional experiments are needed.
- Rev#3 makes good technical points that we editorially find valid. Please try to address these concerns as much as possible. We do not find point #4 (different concentrations of drugs or time series) necessary for publication, provided that the measured effects at the presently used concentrations are all evaluated by proper statistical testing (this may be missing in Figure 2E, and in 3C the $p=0.075$ value seems potentially borderline?). Point #5 seems to be potentially addressed in figure 3C, but a statistical test seems to be missing.

Please let us know if you anticipate any issues tackling the reviewers' points or would like to discuss the revision further. We look forward to your resubmission and would be happy to discuss or help should you have any questions.

GENERAL GUIDELINES:

Text limits: Character count for an Article is < 40,000, not including spaces. Count includes title page, abstract, introduction, results, discussion, acknowledgments, and figure legends. Count does not include materials and methods, references, tables, or supplemental legends.

Figures: Articles may have up to 10 main text figures. Figures must be prepared according to the policies outlined in our Instructions to Authors, under Data Presentation, <http://jcb.rupress.org/site/misc/ifora.xhtml>. All figures in accepted manuscripts will be screened prior to publication.

IMPORTANT: It is JCB policy that if requested, original data images must be made available. Failure to provide original images upon request will result in unavoidable delays in publication. Please ensure that you have access to all original microscopy and blot data images before submitting your revision.

Supplemental information: There are strict limits on the allowable amount of supplemental data. Articles may have up to 5 supplemental figures. Up to 10 supplemental videos or flash animations are allowed. A summary of all supplemental material should appear at the end of the Materials and methods section.

The typical timeframe for revisions is three months; if submitted within this timeframe, novelty will not be reassessed at the final decision. Please note that papers are generally considered through only one revision cycle, so any revised manuscript will likely be either accepted or rejected.

Thank you for this interesting contribution to Journal of Cell Biology. You can contact us at the journal office with any questions, cellbio@rockefeller.edu or call (212) 327-8588.

Sincerely,

Bas van Steensel, PhD
Monitoring Editor, Journal of Cell Biology

Melina Casadio, PhD
Senior Scientific Editor, Journal of Cell Biology

Reviewer #1 (Comments to the Authors (Required)):

In this study Nagashima et al. have applied single nucleosome imaging to examine chromatin dynamics in living cells. The technical side of nucleosome labeling is based on previous studies, which are now used to demonstrate that the transcriptional activity of RNA Pol II has a significant

influence on the degree of chromatin motion. Using a series of experimental approaches the authors nicely show that disrupting Pol II activity leads to an increase in chromatin motion. In other words, the main finding of this study is that Pol II is pivotal to the constraining of genome dynamics. Altogether, this is an interesting and important study - the data are clearly presented and convincing - and in my opinion is suitable for the Journal of Cell Biology. Following are comments that I hope will help improve the manuscript:

- * Fig. 1 vs Fig. 1S - the levels of tagged H2B in fig. 1 are lower than endogenous, but in S1 it looks different. Does this have any implication?
- * Please clarify in the legend what is depicted in Fig. 1D.
- * So that cell biologists that are unfamiliar with MSD analysis understand what they see in the plots (Fig. 1H, 1I & S1E) the authors should provide some explanation to clarify these terms, which come from the biophysics world. Probably there should be a reference to a paper that explains how MSD is interpreted. I think the same goes for explaining the single molecule assay in Fig. 1F (and for this - how many times this was performed, on how many molecules, note in the plot what is grey and black). And an explanation is needed also for the OI-DIC microscopy.
- * Fig. S1B - what is the picture under the plot?
- * Fig. 1G - would be nice to see the tracking in the movie, or on an enlarged section of the movie.
- * Fig. 2 - It's really hard to see the punctate Pol II staining. Maybe add an enlargement.
- * For sake of reproducibility I think it would be useful to be more specific to which Abs are described in the sentence "Immunostaining of two active RNAPII markers, phosphorylated serine 5 (Ser5P) and serine 2 (Ser2P) of CTD (Stasevich et al., 2014)" - actually Kimura used Abs from another study that tested a whole range of anti-P-Ser-CTD Abs. So, were the antibodies used here the famous H5 and H14 Abs? I don't think the actual information is listed in the Methods section.
- * Estimated Rc of nucleosome motion - I suppose there should be a range (+/-) and some statistics to show whether there is a significant change when conditions are perturbed.
- * Fig. 3 - ActD - this inhibitor can be used at much lower concentrations and then only affect Pol I in the nucleolus. I wonder if the authors tried these conditions. What does CX-5461 do to the RNA polymerase and its phosphorylation?
- * Fig. S3C - isn't there a missing picture of the signal in the periphery?
- * Fig. 4C - what is in each lane?
- * Fig. 4D - what is P? The bands in 4D look different than those in S4A.
- * Comparing the plot of the untreated conditions in Fig. 2C to 4F - in 4F untreated cells the MSD looks like the treated cells in 2C.
- * Comparing the images of 3 day starvation in 5C vs 5D - 5C seems more dramatic.
- * What is the y-axis in 5E?
- * Can the authors rescue the Pol II (wash out Auxin?) and then see what levels of Pol II are necessary to return to normal chromatin dynamics.
- * Fig. 6 - an option that must be considered is that the UV is causing DNA breaks and this is leading to the change in DNA dynamics, which is unrelated to the polymerase. A study by David Bazett-Jones showed that such a treatment led to the increased motion of PML bodies due to damage to DNA integrity.
- * I didn't manage to play the two final movies
- * Discussion - "While the classical transcription factory model is consistent with our finding..." - They then provide details of recent studies that are less keen on the transcription factory model. As a reader, I wondered what their opinion is in this debate, but the authors seem to be a little diplomatic. Maybe they would like to be more specific?
- * Discussion - "using a very elegant fluorescent labeling of transcriptional regulatory elements, Gu et al. demonstrated that their movements indeed increased upon transcription activation (Gu et al., 2018)." There are two studies that reach quite different conclusions regarding gene motion due to transcriptional activity - Gu et al 2018 and Germier et al. 2017. In light of the study now presented in

this manuscript, I think need some more elaboration is required in the Discussion

* Statistics are missing in some of the plots.

* There are many grammatical errors, to list a few:

Page 4 - "form loose network"  form a loose network

Page 5 - "treated with various transcription inhibitions"  inhibitors

Page 13 - "We inferred that inhibition or removal of RNAPII-Ser5P can lose network connections"  loosen

Reviewer #2 (Comments to the Authors (Required)):

The manuscript by Nagashima and colleagues describes an experiment using H2B labeling to measure genome mobility in response to a number of drug challenges and compare these changes to the changes in mobility they find correlated to an increase of transcription.

This is an excellent manuscript, likely among the very best I have reviewed this year. The experiments are laid out clearly, the controls are reasonable and use a wide variety of supporting methods. The cell line chosen is suitable for the experiment and the logic in the arguments being presented is clear.

From the beginning the authors related their findings to the concept of transcription factories as suggested by Cook. The authors emphasize that these factories would be dynamic. What remains unclear to me is this: both the labeling and challenges (whether drugs, genetic manipulation, nuclear location) are 'global'. What I mean by this is that some H2B are labeled to get single molecule conditions, but where they are located, on which chromosome or part of a gene, is unknown. Similarly the challenges clearly have an impact, but if that is right next to, close or somewhat distance to a labeled H2B is also unknown. The effect of loss of mobility with increased transcription (or vice versa) can be caused certainly by the suggested formation of transcription hubs, but could similarly be a result of local stiffness. The simulation that is presented, in my understanding, supports the hub, but doesn't rule out other polymer typical reasons such as changes in stiffness. This limit seems not to be discussed.

The second comment I have is that the idea of transcription factories is introduced, but the extent of how many genes would form such a factory is not addressed. It seems for this reason the authors introduce the concept of a hub, which could possibly result from the transcription of a single gene through interaction of several distributed regulatory elements. Once the hub is introduced it is used synonymously with factory. This is confusing as the data are, correctly, interpreted as averages. While one could argue that due to the sparse labeling of H2B (which is the right thing to do) formation of larger clusters might not be detected measuring a large enough number of cells should overcome this limit. My point is: this experiment does not show the formation of factories and is likely not meant to do so, but the synonymous use of the hub concept and the factory idea seems to imply this is the case. I agree with all the analysis, but feel this point is left a bit unsharp. With a global labeling and challenge approach it is hard to interpret the data as far reaching as is implied by the authors.

Other than that I only have some minor observations:

1) On page 7 the constraint radius for living cells and fixed cells is presented, but it is unclear what causes a 56 nm radius (or a >100nm distance) of mobility in fixed cells. One possible explanation could be that 56 nm is in the range of the localization precision of the experiment (low power, live

cells, ...). Based on the meticulous work of the authors in general I would expect they can provide some reasoning or explanation for this in the methods section.

2) On page 8 the term: "longer time window" is used for a MSD measurement. I missed this being introduced, but assume it refers to the number of frames that are allowed between distance measures?

3) I have only reviewed the imaging part of the Materials and Methods section. It is unclear, although likely, if z-stacks were recorded and are the entity that is projected. It is clear that the authors use a 2D analysis and expect homogenous properties in the nucleus for the 3D dimension. Is this approach used to reduce exposure of cells to light? What are the integration times used on the delta vision, what filters and camera were used? How much power is applied?

4) For the density estimation it is hard to understand what was done exactly without the ImageJ scripts. For better reproducibility it would be best if scripts would be published as supporting material.

5) For the single H2B imaging it is stated in the manuscript that HALO illumination iOS used, in the methods section it is unclear though if the necessary iris in the conjugated plane is existing and what beam with was chosen. Makio Tokunaga explained this in great detail. From the methods description it seems though as if a TIRF system was aligned to an oblique angle only.

6) The analysis software is likely MetaMorph and not MataMorph

7) No power settings are given for the H2B imaging.

8) Is ImageJ or ImageJ 2/FIJI used? They should be cited accordingly.

Reviewer #3 (Comments to the Authors (Required)):

In their manuscript, Nagashima et al. imaged single nucleosome movement by tracking TMR tagged H2B-Halo in RPE-1 and colorectal cancer cell lines, and measured nucleosome dynamics under various drug interferences and physiological conditions. The authors observed a positive correlation between RNA polymerase II (PolII) activity and the global chromatin mobility. Furthermore, the authors linked their observations to a model proposing that polII clustering induces a loose chromatin network. While the physiological and genetic perturbations performed in this manuscript are thorough and impressive, there are some serious issues (in order of importance):

1. The authors used oblique illumination to obtain single particle trajectories. By assuming isotropic movement, the measured 2D MSD are extrapolated to 3D. This could lead to biased sampling especially when measuring at the nuclear periphery. From looking at the raw data (Supp Fig. 1 F-G), particle segmentation in SuppFig 1F is not convincing. Clearly not all particles identified in SuppFig 1G (left panel) are well-separated diffraction-limited objects. This segmentation issue could result in poor tracking quality and therefore large errors in the calculation of high order moments. For example, if we compare the alpha-AM treatment data in Fig2C and Fig3C where the same measurements seem to be repeated, are they statistically different or not? The authors need to work a bit harder to convince the reader that all these issues have been thought through, how they have been dealt with, and to what extent they influence the conclusions of the paper.

2. Previous work focusing on the relationship between transcription and chromatin dynamics was able to mark specific chromatin segments and to simultaneously measure transcription activity and chromatin movement (Germier et al 2017 28978433, Gu et al 2018 29371426, Chen et al 2018 30038397). Unlike these studies, the current manuscript measures nucleosome movement without markers for transcriptional activity at specific loci. Since the way RNA PolII activity influence nucleosome positioning is an unresolved question, interpreting nucleosome tracking results as

chromatin behavior seems a bit adventurous. Along the same line of thought, the chromatin network model the authors propose lacks direct experimental evidence. For example, can any kind of spatial correlation of chromatin dynamics be determined from the data?

3. The authors mentioned that "the MSD plots were well fitted to a subdiffusion model". What is this model? What is the scaling power and diffusion coefficient? A log-log plot should be reported. This diffusion model is not trivial because the authors discuss physical constraint all throughout the manuscript. How does the physical constraint fit into their subdiffusion model?

4. Is there any dosage effect of the drugs used in the study? Is it possible to use different concentrations of DRB or alpha-AM? Or is it possible to tune the auxin level that induces different efficiency of the AID system? Another approach would be to image time series, i.e. measuring MSD from the SAME cells at a different time point after drug treatment. These approaches are necessary to make the correlation revealed by the authors more convincing.

5. Is there any spatial heterogeneity in the time-averaged MSD calculated from single trajectories? For example in Fig2D, does the alpha-AM treatment influence the nucleosomes in the middle of the nucleus more than it does to the nucleosomes at the periphery? Also the numbers of trajectories used for calculating the ensemble-averaged MSD for each cell and their length distributions should be reported.

6. What is the 'P' lane on the western blot in Fig 4D? The parent line for the transfections? This should be mentioned in the figure caption. Why do the blotting patterns in the 'P' lane and in the dox- auxin- controls look different? The whole immunoblotting picture should be provided.

7. Does the model capture the whole distribution (not just the mean) of the time-averaged MSD calculated from individual trajectories?

8. Figure 5E needs y-axis label.

9. In the Methods section 'Single nucleosome imaging microscopy', the detection range is likely to be '575-710 nm', not '75-710 nm'?

10. What is the image under SuppFig1B?

Response to the Editor's and Reviewers' Comments

We would like to thank the Editors and Reviewers for their thoughtful and insightful comments. We believe that we have improved the quality of our manuscript by constructively responding to their suggestions and comments. Our point-by-point responses are provided below.

Reviewer #1 (Comments to the Authors (Required)):

In this study Nagashima et al. have applied single nucleosome imaging to examine chromatin dynamics in living cells. The technical side of nucleosome labeling is based on previous studies, which are now used to demonstrate that the transcriptional activity of RNA Pol II has a significant influence on the degree of chromatin motion. Using a series of experimental approaches the authors nicely show that disrupting Pol II activity leads to an increase in chromatin motion. In other words, the main finding of this study is that Pol II is pivotal to the constraining of genome dynamics. Altogether, this is an interesting and important study - the data are clearly presented and convincing - and in my opinion is suitable for the Journal of Cell Biology. Following are comments that I hope will help improve the manuscript:

We would like to thank Reviewer#1 for his/her supportive words regarding our work.

** Fig. 1 vs Fig. 1S - the levels of tagged H2B in fig. 1 are lower than endogenous, but in S1 it looks different. Does this have any implication?*

The two experiments had a different purpose: In Fig. 1A, we showed how much H2B-Halo was expressed in the cells as compared to endogenous H2B (i.e. expression of H2B-Halo is about 5-10% of that of endogenous H2B). On the other hand, in Fig. S1A, to reliably detect behavior of H2B-Halo during a salt extraction, we took a longer exposure condition for H2B-Halo detection in western blotting. We noted this in the Figure S1A legend.

** Please clarify in the legend what is depicted in Fig. 1D.*

A proper explanation was added in the Fig. 1D legend.

** So that cell biologists that are unfamiliar with MSD analysis understand what they see in the plots (Fig. 1H, II & S1E) the authors should provide some explanation to clarify these terms, which come from the biophysics world. Probably there should be a reference to a paper that explains how MSD is interpreted. I think the same goes for explaining the single molecule assay in Fig. 1F (and for this - how many times this was performed, on how many molecules, note in the plot what is grey and black). And an explanation is needed also for the OI-DIC microscopy.*

We would like to thank Reviewer #1 for raising these important points. We added a proper brief explanation and reference for MSD. For OI-DIC imaging, a scheme to help readers' understanding was added to Fig. S2E. Regarding the old Fig. 1F, this is a control experiment showing by single-step photobleaching that each dot represents a single molecule, and is not a main analysis of single nucleosome imaging. To avoid confusion, the old Fig. 1F was transferred to Fig. S1B.

** Fig. S1B - what is the picture under the plot?*

The old Fig. S1B is a point spread function (PSF) of a single nucleosome dot, which shows a horizontal view of how long the dot was optically stretched in z-direction. We added this explanation in the new Fig. S1D.

** Fig. 1G - would be nice to see the tracking in the movie, or on an enlarged section of the movie.*

According to this suggestion, we added a tracking movie as Video2.

** Fig. 2 - It's really hard to see the punctate Pol II staining. Maybe add an enlargement.*

We added an enlarged Ser5P-RNAPII staining in the new Fig. 2C.

** For sake of reproducibility I think it would be useful to be more specific to which Abs are described in the sentence "Immunostaining of two active RNAPII markers, phosphorylated serine 5 (Ser5P) and serine 2 (Ser2P) of CTD (Stasevich et al., 2014)" - actually Kimura used Abs from another study that tested a whole range of anti-P-Ser-CTD Abs. So, were the antibodies used here the famous H5 and H14 Abs? I don't think the actual information is listed in the Methods section.*

We listed additional detailed information on the antibodies in Material and methods.

** Estimated Rc of nucleosome motion - I suppose there should be a range (+/-) and some statistics to show whether there is a significant change when conditions are perturbed.*

We added a standard deviation of Rc in the new Fig. 1H legend.

** Fig. 3 - ActD - this inhibitor can be used at much lower concentrations and then only affect Pol I in the nucleolus. I wonder if the authors tried these conditions. What does CX-5461 do to the RNA polymerase and its phosphorylation?*

We agree to Reviewer#1's point. We added MSD data with low concentration of ActD (Fig. S2C), which was effective against RNA PolI but not RNA PolIII. In this condition,

ActD did not affect chromatin motion (Fig. S2C). For CX-treatment, we did not see any significant effect on Ser5P-RNAPII and the data was included in the new Fig. 4B.

** Fig. S3C - isn't there a missing picture of the signal in the periphery?*

In the nuclear periphery, signals of Ser5P-RNAPII are very low. To better demonstrate this point, we made a merged image of DAPI and Ser5P-RNAPII images with their intensity line plots (right, new Fig. 2C).

** Fig. 4C - what is in each lane?*

We expanded the Figure legend (the new Fig. 5C) for the old Fig. 4C.

** Fig. 4D - what is P? The bands in 4D look different than those in S4A.*

“P” meant “parental cells”. We improved the label of the panels (the new Fig. 5C and D). An explanation was provided in the new Fig. S3A legend.

** Comparing the plot of the untreated conditions in Fig. 2C to 4F - in 4F untreated cells the MSD looks like the treated cells in 2C.*

We agree with Reviewer#1 that the nucleosome movements of DLD1 and RPE-1 cells are somehow different: DLD1 cells have higher MSD value. We found this interesting as a future direction. That DLD1 cells have a somehow higher MSD value than RPE-1 cells was noted in the new Fig. 5F legend.

** Comparing the images of 3 day starvation in 5C vs 5D - 5C seems more dramatic.*

We understand Reviewer#1's point. Starvation of the old Fig. 5D (new Fig. 6D) might be slightly weaker than that of the old Fig. 5C (new Fig. 6C). But please note that there is still a significant difference between Ser5P-RNAPII signal intensities of 0 day and 3-day starvation in the old Fig. 5D (new Fig. 6D).

** What is the y-axis in 5E?*

The y-axis in the old Figure 5E (new Fig. 6E) shows MSD and this information was added to the graph.

** Can the authors rescue the Pol II (wash out Auxin?) and then see what levels of Pol II are necessary to return to normal chromatin dynamics.*

Following Reviewer#1's suggestion, we performed the suggested rescue experiment of AID-RNAPII by washing out auxin, and observed that chromatin dynamics returned to the normal level 12 hours after removing auxin. The data was included in the new Fig. 5F.

** Fig. 6 - an option that must be considered is that the UV is causing DNA breaks and this is leading to the change in DNA dynamics, which is unrelated to the polymerase. A study by David Bazett-Jones showed that such a treatment led to the increased motion of PML bodies due to damage to DNA integrity.*

We agree to this suggested point. We mentioned the possibility and cited a proper reference by Bazett-Jones in the paper:

“although we cannot fully rule out the possibility that DNA damage also contributes to chromatin decondensation and subsequent increase in its motion (Dellaire et al., 2006).”

** I didn't manage to play the two final movies*

We reformatted the two movies in avi format (Videos 5 and 6) and checked them on both PC and Mac platforms.

** Discussion - "While the classical transcription factory model is consistent with our finding..." - They then provide details of recent studies that are less keen on the transcription factory model. As a reader, I wondered what their opinion is in this debate, but the authors seem to be a little diplomatic. Maybe they would like to be more specific?*

We agree with Reviewer#1. Our data seems to be compatible with the classical transcription factory model and more recent clustering/droplet formation of transcription related factors by phase separation. To make this point clear, we extensively improved the related Discussion part.

** Discussion - "using a very elegant fluorescent labeling of transcriptional regulatory elements, Gu et al. demonstrated that their movements indeed increased upon transcription activation (Gu et al., 2018)." There are two studies that reach quite different conclusions regarding gene motion due to transcriptional activity - Gu et al 2018 and Germier et al. 2017. In light of the study now presented in this manuscript, I think need some more elaboration is required in the Discussion.*

Following the suggestions by Reviewer#1, as well as Reviewers#2 and #3, we mentioned the three papers: Gu et al. and Germier et al. demonstrated that their movements indeed increased upon transcription activation (Gu et al., 2018; Germier et al. 2017) while Chen et al. observed an opposite effect (Chen et al., 2018), which is similar to ours.

** Statistics are missing in some of the plots.*

We added statistics to the plots that lacked it.

** There are many grammatical errors, to list a few:*

Page 4 - "form loose network"  form a loose network

Page 5 - "treated with various transcription inhibitions"  inhibitors

Page 13 - "We inferred that inhibition or removal of RNAPII-Ser5P can lose network connections"  loosen

We corrected them and the manuscript was checked by a native English speaker.

Reviewer #2 (Comments to the Authors (Required)):

This is an excellent manuscript, likely among the very best I have reviewed this year. The experiments are laid out clearly, the controls are reasonable and use a wide variety of supporting methods. The cell line chosen is suitable for the experiment and the logic in the arguments being presented is clear.

We would like to thank Reviewer#2 so much for his/her supportive and encouraging words on our work.

From the beginning the authors related their findings to the concept of transcription factories as suggested by Cook. The authors emphasize that these factories would be dynamic. What remains unclear to me is this: both the labeling and challenges (whether drugs, genetic manipulation, nuclear location) are 'global'. What I mean by this is that some H2B are labeled to get single molecule conditions, but where they are located, on which chromosome or part of a gene, is unknown.

We understand Reviewer#2's point. There are some previous interesting studies focusing on chromatin dynamics of a single locus. We would stress that while the location (chromosome or gene) of our labeled nucleosomes is unknown, a global view revealed an interesting link between chromatin organization and dynamics: loose genome chromatin networks via active RNA polymerase II. We believe that seeing not only the individual "tree" (genome locus) but also the "forest" (genome wide chromatin) is important to understand nature of chromatin organization in the living cells.

Similarly the challenges clearly have an impact, but if that is right next to, close or somewhat distance to a labeled H2B is also unknown. The effect of loss of mobility with increased transcription (or vice versa) can be caused certainly by the suggested formation of transcription hubs, but could similarly be a result of local stiffness. The simulation that

is presented, in my understanding, supports the hub, but doesn't rule out other polymer typical reasons such as changes in stiffness. This limit seems not to be discussed.

The second comment I have is that the idea of transcription factories is introduced, but the extent of how many genes would form such a factory is not addressed. It seems for this reason the authors introduce the concept of a hub, which could possibly result from the transcription of a single gene through interaction of several distributed regulatory elements. Once the hub is introduced it is used synonymously with factory. This is confusing as the data are, correctly, interpreted as averages. While one could argue that due to the sparse labeling of H2B (which is the right thing to do) formation of larger clusters might not be detected measuring a large enough number of cells should overcome this limit. My point is: this experiment does not show the formation of factories and is likely not meant to do so, but the synonymous use of the hub concept and the factory idea seems to imply this is the case. I agree with all the analysis, but feel this point is left a bit unsharp. With a global labeling and challenge approach it is hard to interpret the data as far reaching as is implied by the authors.

We would appreciate Reviewer#2 (also Reviewer#3) for raising these important points. We agree that we cannot rule out the raised possibility that the effect of loss of mobility with increased transcription (or vice versa) could be a result of local stiffness upon RNAPII-binding. While we mentioned the possibility, we consider it is unlikely. The reasons were discussed in Discussion.

Briefly, first, our model (Fig. 8A) that P-TEFb clusters and RNAPII-Ser5P are a hub and glue, respectively, is supported by the perturbations of P-TEFb clusters by DRB (Fig. 3A) or CDK9 KD (Fig. 8B), or removal of RNAPII (glue) (Fig. 5F; Fig. S3C), which can potentially remove connections between the hubs and chromatin domains. Additionally, our computational simulation result is consistent with our model. Second, available data suggest that RNAPII-Ser5P is involved in connecting chromatin domains, but not in chromatin domain formation (center, Fig. 8A). Finally, our model is compatible with the classical transcription factory model (Buckley and Lis, 2014; Feuerborn and Cook, 2015) and with recent reports that RNAPII and other factors form dynamic clusters/droplets, presumably by a phase separation process (Boehning et al., 2018; Boija et al., 2018; Cho et al., 2018; Chong et al., 2018; Lu et al., 2018; Sabari et al., 2018).

We would also emphasize that formation of transcription hubs, which constrain chromatin domains, can explain why active RNAPII, which locates to only limited regions of the genome, can globally stabilize chromatin motion.

To address the second comment, in addition to the points described above, we would mention our previous finding that active RNAPII clusters were often localized outside of the chromatin domains (Nozaki et al., 2017; Markaki et al., 2010; Xu et al., 2018). Taken together, we suggested loose genome chromatin networks via active RNA polymerase II.

Other than that I only have some minor observations:

1) On page 7 the constraint radius for living cells and fixed cells is presented, but it is unclear what causes a 56 nm radius (or a >100nm distance) of mobility in fixed cells. One possible explanation could be that 56 nm is in the range of the localization precision of the experiment (low power, live cells, ...). Based on the meticulous work of the authors in general I would expect they can provide some reasoning or explanation for this in the methods section.

We agree to Reviewer#2's impression. To address this comment, we added standard deviation of R_c values and their statistical analysis, showing that the nucleosome R_c of living cells are significantly greater than that of FA-fixed cells. On the other hand, the result suggests that the chromatin in FA-fixed cells still has considerable local movements, presumably because FA is a rather weak fixative. This property might be important for cell biology experiments, for instance to ensure the antibody penetration into its target in the FA-fixed cells.

2) On page 8 the term: "longer time window" is used for a MSD measurement. I missed this being introduced, but assume it refers to the number of frames that are allowed between distance measures?

The "longer time window" meant longer time tracking for each dot. Instead of the usual 0.5 sec tracking time, 3 seconds was used for tracking time. We made this point clear in the new Figure 1H legend.

3) I have only reviewed the imaging part of the Materials and Methods section. It is unclear, although likely, if z-stacks were recorded and are the entity that is projected. It is clear that the authors use a 2D analysis and expect homogenous properties in the nucleus for the 3D dimension. Is this approach used to reduce exposure of cells to light? What are the integration times used on the delta vision, what filters and camera were used? How much power is applied?

We agree with Reviewer#2 that "Quantification of immunostaining images" section in the Materials and methods did not provide enough information. We added more specific information for DeltaVision microscopy and imaging conditions. Image projections were done because the signals were not distributed homogeneously across z-stacks.

4) For the density estimation it is hard to understand what was done exactly without the ImageJ scripts. For better reproducibility it would be best if scripts would be published as supporting material.

How to measure total material density in the living cells was reported recently by us (Imai et al. 2016). Although the estimation procedure is not so simple to write it in a paragraph, we provided a scheme in Fig. S2E and more information in the Materials and methods section.

5) For the single H2B imaging it is stated in the manuscript that HALO illumination iOS used, in the methods section it is unclear though if the necessary iris in the conjugated plane is existing and what beam with was chosen. Makio Tokunaga explained this in great detail. From the methods description it seems though as if a TIRF system was aligned to an oblique angle only.

TIRF system by NIKON was aligned to an oblique angle without setting an iris in the conjugated plane.

6) The analysis software is likely MetaMorph and not MataMorph

We corrected it.

7) No power settings are given for the H2B imaging.

We added the proper information in the Materials and methods section. Laser power coming out from the objective was about 5 mW.

8) Is ImageJ or ImageJ 2/FIJI used? They should be cited accordingly.

Fiji was used and we cited it properly.

Reviewer #3 (Comments to the Authors (Required)):

1. The authors used oblique illumination to obtain single particle trajectories. By assuming isotropic movement, the measured 2D MSD are extrapolated to 3D. This could lead to biased sampling especially when measuring at the nuclear periphery. From looking at the raw data (Supp Fig. 1 F-G), particle segmentation in SuppFig 1F is not convincing. Clearly not all particles identified in SuppFig 1G (left panel) are well-separated diffraction-limited objects. This segmentation issue could result in poor tracking quality and therefore large errors in the calculation of high order moments. For example, if we compare the alpha-AM treatment data in Fig2C and Fig3C where the same measurements seem to be repeated, are they statistically different or not? The authors need to work a bit harder to convince the reader that all these issues have been thought through, how they have been dealt with, and to what extent they influence the conclusions of the paper.

We would appreciate Reviewer#3 for raising some critical issues for readers' better understanding.

First, regarding assuming isotropic movement, at the nuclear periphery, where the nucleosome movements can be constrained by nuclear lamina, it is possible that the nucleosome movements in z-direction are more constrained than those in x-y direction. Therefore, our corrected MSD in 3D at the nuclear periphery might be overestimated, as Reviewer#3 might be concerned. However, considering that the obtained MSD value is still significantly smaller than that in the nuclear interior, this issue is not so problematic.

Second, the old Supplemental Fig. 1F-G were not raw data, but rather panels to explain the “chromatin heat map” visualizing spatial chromatin dynamics in the nucleus. To avoid confusion, the old Supplemental Fig. 1F was deleted and the explanation was improved in the new Fig. S1H legend.

Third, no significant difference (p value = 0.1291) was found between nucleosome motions in the α -AM treated cells in the old Fig. 2C (new Fig. 3A) and Fig. 3C (new Fig. 4F).

Finally, for better understanding of the nucleosome particle tracking process, some tracking examples were presented as Video2.

2. Previous work focusing on the relationship between transcription and chromatin dynamics was able to mark specific chromatin segments and to simultaneously measure transcription activity and chromatin movement (Germier et al 2017 28978433, Gu et al 2018 29371426, Chen et al 2018 30038397). Unlike these studies, the current manuscript measures nucleosome movement without markers for transcriptional activity at specific loci. Since the way RNA PolII activity influence nucleosome positioning is an unresolved question, interpreting nucleosome tracking results as chromatin behavior seems a bit adventurous. Along the same line of thought, the chromatin network model the authors propose lacks direct experimental evidence. For example, can any kind of spatial correlation of chromatin dynamics be determined from the data?

We thank Reviewer#3, as well as Reviewer#2, for raising these important points.

We agree with Reviewer#3 that how RNAPII influences nucleosome movements is not so clear. However, as we addressed Reviewer#2's comment, we would stress that formation of transcription hubs, which constrain chromatin domains, can well explain why active RNAPII, which locates only at limited regions of the genome, can globally stabilize chromatin motion.

The Discussion part was extensively reorganized:

Briefly, first, our model (Fig. 8A) that P-TEFb clusters and RNAPII-Ser5P are a hub and glue, respectively, is supported by the perturbations of P-TEFb clusters by DRB (Fig. 3A) or CDK9 KD (Fig. 8B), or removal of RNAPII (glue) (Fig. 5F; Fig. S3C), which can potentially remove connections between the hubs and chromatin domains. Besides, our

computational simulation result is consistent with our model. Second, available data suggest that RNAPII-Ser5P is involved in connecting chromatin domains, but not in chromatin domain formation (center, Fig. 8A). Finally, our model is compatible with the classical transcription factory model (Buckley and Lis, 2014; Feuerborn and Cook, 2015) and with recent reports that RNAPII and other factors form dynamic clusters/droplets, presumably by a phase separation process (Boehning et al., 2018; Boija et al., 2018; Cho et al., 2018; Chong et al., 2018; Lu et al., 2018; Sabari et al., 2018).

In our paper, we would also emphasize importance of seeing not only the individual “tree” (genome locus) but also the “forest” (genome wide chromatin) to understand nature of chromatin organization in the living cells. This point was included in Discussion.

Regarding spatial correlation of chromatin dynamics, we agree that it is an important next issue in our study. Actually a spatial correlation of chromatin movements upon transcription was recently suggested {Shaban, 2018; Pubid 30038397} and we mentioned it in Discussion.

3. The authors mentioned that "the MSD plots were well fitted to a subdiffusion model". What is this model? What is the scaling power and diffusion coefficient? A log-log plot should be reported. This diffusion model is not trivial because the authors discuss physical constraint all throughout the manuscript. How does the physical constraint fit into their subdiffusion model?

To address this comment, we examined the scaling power and diffusion coefficient of the plots corresponding to the untreated and α -AM treated cells. They were added to the new Figs. 1H and 3C legends.

In addition, since we agree that “a subdiffusion model” is not appropriate phrase, it was changed into “subdiffusive”.

4. Is there any dosage effect of the drugs used in the study? Is it possible to use different concentrations of DRB or alpha-AM? Or is it possible to tune the auxin level that induces different efficiency of the AID system? Another approach would be to image time series, i.e. measuring MSD from the SAME cells at a different time point after drug treatment. These approaches are necessary to make the correlation revealed by the authors more convincing.

We agree with Reviewer#3 that it is important to show the dose-dependency of these drugs. We added data on α -AM-, DRB- and ActD-treated cells with a low dose in the new Fig. S2C.

5. Is there any spatial heterogeneity in the time-averaged MSD calculated from single trajectories? For example in Fig2D, does the alpha-AM treatment influence the

nucleosomes in the middle of the nucleus more than it does to the nucleosomes at the periphery? Also the numbers of trajectories used for calculating the ensemble-averaged MSD for each cell and their length distributions should be reported.

Yes, as we showed in the new Fig. 3B, there is a considerable spatial heterogeneity in MSD. α -AM treatment was not effective at the nuclear periphery (new Fig. 4F). The numbers of trajectories used for calculating the ensemble-averaged MSD for each cell is about 1000. The requested distributions were already shown as “chromatin heat map” (new Fig. 3B).

6. What is the 'P' lane on the western blot in Fig 4D? The parent line for the transfections? This should be mentioned in the figure caption. Why do the blotting patterns in the 'P' lane and in the dox- auxin- controls look different? The whole immunoblotting picture should be provided.

We would appreciate Reviewer#3 for raising important points. Yes, “P” lane is for the parent line. We added it to the Fig. 5 C and D legends.

Regarding the second question, we agree that the blotting patterns of RPB1 of RNAPII appear to be different between the 'P' and the Dox- Auxin- controls. This is because RPB1 protein was fused with AID-tag and fluorescent protein mClover (totally ~35 kDa) in the AID cells and consequently the cells have larger RPB. We noted this point in the Fig. 5D legend.

7. Does the model capture the whole distribution (not just the mean) of the time-averaged MSD calculated from individual trajectories?

We agree to Reviewer#3's point. Indeed, quantitative comparison with the nucleosome movements in living cells would be a next interesting issue. For our computational model to show more quantitatively, the standard deviation, scaling power and diffusion coefficient of the model were examined and shown. Furthermore, we added MSD distribution of the models with and without RNAPII. Indeed, this distribution implies how the transcription machinery including RNAPII can constrain global chromatin motion: There are slow RNAPII-bound chromatin fractions (slow peak) and fast remaining chromatin fractions (fast peak), which lead to the reduction of averaged chromatin motion. This point was described in the Fig. 9C legend and Discussion, and provides an important clue for further analysis of nucleosome movements in living cells.

8. Figure 5E needs y-axis label.

We added the label.

9. In the Methods session 'Single nucleosome imaging microscopy', the detection range is likely to be '575-710 nm', not '75-710 nm'?

We corrected it.

10. What is the image under SuppFig1B?

This is a point spread function (PSF) of the single nucleosome, which shows a horizontal view of how long the dot was optically stretched in z-direction. We added this explanation in the Fig. S1D legend.

February 1, 2019

RE: JCB Manuscript #201811090R

Dr. Kazuhiro Maeshima
National Institute of Genetics
Yata 1111
Mishima 411-8540
Japan

Dear Dr. Maeshima,

Thank you for submitting your revised manuscript entitled "Single nucleosome imaging reveals loose genome chromatin networks via active RNA polymerase II". Thank you for the numerous clarifications and additions to resolve the reviewers' concerns. We have assessed your rebuttal and revised manuscript and feel that the revisions are reasonable and adequate. We would be happy to publish your paper in JCB pending final revisions necessary to meet our formatting guidelines (see details below).

1) Titles: Please consider the following revision suggestions aimed at increasing the accessibility of the work for a broad audience and non-experts.

Running title (50 characters max, including spaces -- the added characters are OK and we can edit this for you in our system if you run into technical issues due to the character count): Active RNAPII globally constrains chromatin movements

2) Materials and methods: Should be comprehensive and not simply reference a previous publication for details on how an experiment was performed. Please provide full descriptions in the text for readers who may not have access to referenced manuscripts.

- Please provide a brief description of all genetic constructs and cell lines, even if described in other work (e.g.: Nup107-Venus expressed HeLa cells)

- Please indicate the species for all antibodies

- Please include the sequences for all RNAi oligos, even negative controls, if they were made available to you from the manufacturer

- We encourage you to make your Image analysis code available (as suggested by Reviewer #2, point 4) in this paper as well (it can be hosted as a supplemental material, typically as a txt file)

- Microscope image acquisition: The following information must be provided about the acquisition and processing of images:

a. Make and model of microscope

b. Type, magnification, and numerical aperture of the objective lenses

c. Temperature

d. imaging medium

e. Fluorochromes

f. Camera make and model

g. Acquisition software

h. Any software used for image processing subsequent to data acquisition. Please include details and types of operations involved (e.g., type of deconvolution, 3D reconstitutions, surface or volume rendering, gamma adjustments, etc.).

3) References: There is no limit to the number of references cited in a manuscript. References should be cited parenthetically in the text by author and year of publication.

- Please note our formatting requirements for preprint citations: <http://jcb.rupress.org/reference-guidelines>

Yesbolatova, A., T. Natsume, K. Hayashi, and M.T. Kanemaki. 2019. Generation of conditional auxin inducible degron (AID) cells and tight control of degron-fused proteins using the degradation inhibitor auxinole. bioRxiv.

A. MANUSCRIPT ORGANIZATION AND FORMATTING:

Full guidelines are available on our Instructions for Authors page, <http://jcb.rupress.org/submission-guidelines#revised>. **Submission of a paper that does not conform to JCB guidelines will delay the acceptance of your manuscript.**

B. FINAL FILES:

-- High-resolution figure and video files: See our detailed guidelines for preparing your production-ready images, <http://jcb.rupress.org/fig-vid-guidelines>.

Thank you for this interesting contribution, we look forward to publishing your paper in the Journal

of Cell Biology.

Sincerely,

Bas van Steensel, PhD
Monitoring Editor, Journal of Cell Biology

Melina Casadio, PhD
Senior Scientific Editor, Journal of Cell Biology